# Decentralized Learning with Random Walks and Communication-Efficient Adaptive Optimization

**Aleksei Triastcyn**$^*$
aleksey.tryastsyn@alumni.epfl.ch

**Matthias Reisser**
Qualcomm AI Research
mreisser@qti.qualcomm.com

**Christos Louizos**
Qualcomm AI Research
clouizos@qti.qualcomm.com

## Abstract

We tackle the problem of federated learning (FL) in a peer-to-peer fashion without a central server. While prior work mainly considered gossip-style protocols for learning, our solution is based on random walks. This allows to communicate only to a single peer at a time, thereby reducing the total communication and enabling asynchronous execution. To improve convergence and reduce the need for extensive tuning, we consider an adaptive optimization method – Adam. Two extensions reduce its communication costs: state compression and multiple local updates on each client. We theoretically analyze the convergence behaviour of the proposed algorithm and its modifications in the non-convex setting. We show that our method can achieve performance comparable to centralized FL without communication overhead. Empirical results are reported on a variety of tasks (vision, text), neural network architectures and large-scale federations (up to $\sim 342$k clients).

## 1   Introduction

Federated learning is typically set up with a central server coordinating the training process: selecting clients to participate in each round, sending the current model weights to the clients, and aggregating the received updates. However, in some situations, FL with a central server is impractical due to, for example, costs, privacy concerns, or when peer-to-peer communication (*e.g.*, using Bluetooth) is preferred. This paper tackles the problem of peer-to-peer learning in which users are connected via direct communication links into a potentially sparse communication graph. They perform local updates to the model similarly to FL, but unlike in FL, the decision about what to do next and where to send the model is taken locally (by random selection among clients' neighbors in the graph). One example of the basis for the communication graph is routing times between users; this way, less time can be spent on model transfer and more on training in total.

Existing work has primarily focused on datacenter scenarios and gossip-style protocols for learning [11, 1], [12], [21, 22, 8] where messages are exchanged with all peers of a given client. These methods have been shown to work successfully in data center settings to speed up training [1] and to address unreliable or expensive communication [22]. On the other hand, there is very limited literature on cross-device decentralized FL, and the scale of experiments is impractically small (< 100 clients) [9, 17, 6]. We focus on a different style of algorithms – *random walks* – which allow

---

$^*$Work done while at Qualcomm AI Research. Qualcomm AI Research is an initiative of Qualcomm Technologies, Inc. and/or its subsidiaries.

Workshop on Federated Learning: Recent Advances and New Challenges, in Conjunction with NeurIPS 2022 (FL-NeurIPS'22). This workshop does not have official proceedings and this paper is non-archival.

communicating only to a single peer at a time. They reduce total communication and enable asynchronous execution. While such approaches have been considered before [2, 24, 3, 4], experimental results in FL are similarly limited. We, therefore, aim to close this research gap with our paper.

Along with general learning, we consider the problem of adaptive optimization in the decentralized setting with random walks. While adaptive optimization is straightforward to solve in centralized scenarios by using, for example, Adam [7] on the server side, naively applying it to the random walk setting leads to challenges. The main issue is that the optimizer state (primarily, moments, each of which is equal in size to the model) needs to be transmitted along with the model, which can double or even triple communication costs. We thus propose two strategies to mitigate this effect. Firstly, we propose the use of compression for the optimizer's state. Secondly, we propose performing multiple local updates on each client before communicating to a neighbor. We prove the convergence of such random walks to a vicinity of a stationary point in the non-convex setting and empirically show that they allow for similar communication vs. accuracy tradeoffs to traditional centralized FL. In summary, we make the following contributions in this work:

- we introduce Adam [7], a popular adaptive optimization method, in the decentralized setting with random walks. This allows us to improve performance upon traditional random walk SGD-type of approaches, especially in complex and sparse gradient tasks;
- we introduce two communication-efficient variants of Adam for this specific setting. More specifically, we introduce a novel moment quantization method in the log domain and adopt the multiple local updates paradigm, popularized by [15];
- we provide a theoretical analysis of the convergence behavior of this novel random-walk Adam with our proposed modifications. We show that it has similar rates to what traditional adaptive methods [25] obtain in the centralized setting;
- we perform extensive experiments on various datasets and architectures, with federation sizes ranging from 100 to ~342k clients, showing that our method achieves similar performance to the traditional centralized FedAvg. This is in contrast to prior works, which mostly consider data center scenarios, synthetic data, or a small number of participants (*i.e.*, $< 100$).

## 2  Related Work

Decentralized learning is typically approached from one of two angles: gossip averaging or random walks. Gossip-style algorithms are rather extensively studied in the literature. [11] showed that gossip averaging between computational nodes leads to speedup by eliminating the bottleneck on a central coordinating node (and coincidentally, a central point of failure). [1] presented the stochastic gradient push (SGP) algorithm, which enables asymmetric communication topologies (in contrast to [11]) and therefore circumvents the need for deadlock-avoidance and synchronization. Reliability and convergence speed were further improved by the RelaySGD method by [22] using spanning trees over a communication network to relay model updates between nodes (without decaying their magnitude). In addition to practical advantages over SGP and quasi-global momentum [12], RelaySGD boasts theoretical independence of data heterogeneity (non-iid split between nodes). Since a straightforward implementation of gossip protocols leads to a significant increase in communication, even on sparse graphs, several approaches to compress the transferred updates have been proposed [8, 21]. For example, Choco-SGD and Choco-Gossip by [8], which converge linearly and allow arbitrary compression operators for updates, and PowerGossip [21], which directly compresses the model differences between neighboring nodes using low-rank linear approximation. Importantly, all these works, as well as dedicated FL gossip papers [9, 17, 6], only consider a small number of nodes. For scaled up problems, performance may drop noticeably (see [22, Appendix E.2]).

Algorithms based on random walks received less attention in this context, perhaps due to a potentially high number of communication rounds required for convergence. However, these methods are worth studying since the overall communication may be considerably lower. [2] and [3] presented uniform and weighted random walk SGD algorithms, proved their convergence, and demonstrated a favorable comparison with gossip averaging, but only on synthetic data. [24] proposed inexact stochastic parallel random walk alternating direction method of multipliers (ISPW-ADMM), which utilizes multiple parallel random walks speed up learning, and showed gains over gossiping; however, they only considered 10 devices and it is unclear what kind of data were used. For non-convex problems, [20] studied Markov chain gradient descent, its convergence and mixing rates, while [14] put forth

another ADMM-based random walk algorithm in a decentralized optimization setting. Again, large-scale practical experiments are missing from these works. Therefore, there is a need to compare gossip averaging and centralized FedAvg on realistic ML datasets and with sizeable federations of devices.

Furthermore, adaptive optimization is often preferable to simple SGD due to faster convergence and ease of learning rate tuning. Hence, in contrast to prior work, we study random walks in combination with Adam. To limit the communication overhead from transmitting the optimizer state between clients, we consider compression for moments, similar to [18] and multiple local updates on each node, similar to FedAvg [15]. For moments compression, we look at research on memory-efficient adaptive optimizers. [18] proposed to store per-row and per-column sums of moving averages instead of full matrices for the second moment, using these sums to approximately reconstruct the full moment, and removing the first moment. This approach could be extended to tensors for higher-dimensional convolutional layers, and other factorization and low-rank approximation methods (random projections, power iteration, etc.) could be considered to improve the accuracy of the approximation. Addressing the problem from a different angle, [5] partitioned tensors into separate blocks and quantized them independently (which can be parallelized for efficiency) and achieved almost full-precision performance for 8-bit-per-parameter moments. Finally, count-sketch methods can also be applied [19], reducing memory requirements to logarithmic and having a proven rate of convergence similar to Adam with $\beta_1 = 0$.

## 3 Decentralized Federated Learning via a Random Walk

Assume that we have a pool of $S$ clients where each client $s \in S$ has access to a local dataset $\mathcal{D}_s$ of $N_s$ datapoints. In federated learning (FL), we are interested in learning a vector of model parameters $\mathbf{w}$ according to the following

$$\mathbf{w}^* = \underset{\mathbf{w} \in \mathbb{R}^d}{\arg \min} \, \mathbb{E}_{s \sim p(s)} \left[ \mathbb{E}_{\xi \sim p(\mathcal{D}_s)} \left[ f(\mathbf{w}, \xi) \right] \right], \tag{1}$$

where $p(s) = N_s/N$ and $f(\mathbf{w}, \xi)$ is the loss function computed with parameters $\mathbf{w}$ on a data sample $\xi$. It is easy to see that this corresponds to sampling from the global data distribution (as if data of all clients are pulled together and sampled uniformly at random) to optimize $f(\mathbf{w})$. Traditional FL approaches, such as FedAvg [15], optimize this objective by employing a central coordinator, *i.e.*, the server and exchanging messages in the form of model updates between the clients and the server. In this work, we forego the notion of a central coordinator and instead design an algorithm where the clients can learn a model by communicating amongst themselves in a peer-to-peer fashion.

### 3.1 Decentralized federated learning

Consider a set of clients $S$ connected by a set of communication links $E$ into a graph $\mathcal{G} : (S, E)$, called the *communication graph*. For the rest of this work, we assume that $\mathcal{G}$ is connected, non-bipartite and undirected. Clients can be personal devices, such as smartphones, IoT devices, and so on, as well as organizations (*e.g.*, hospitals, banks, etc.). The connections could be based on physical networks, virtual networks, routing times, trust, or other concepts. Each client $s$, therefore, has a set of neighbors $\mathcal{N}(s) := \{v \in S : (s, v) \in E\}$ defined by the connections available to this client.

---

**Algorithm 1** Basic gradient-based decentralized learning by a random walk

---

Client $s$ receives a message $\mathcal{M}$ from neighbor $h$
$\mathbf{w}^{(t)} \leftarrow \mathcal{M}$
$\mathbf{w}^{(t+1)} \leftarrow \mathbf{w}^{(t)} - \eta \nabla_{\mathbf{w}} f(\mathbf{w}^{(t)}, \mathcal{D}_s^{(t)})$
$j \leftarrow$ random neighbor from $\mathcal{N}(s)$
**return** Send model $\mathbf{w}^{(t+1)}$ to neighbor $j$

---

**Gradient-based learning by a random walk**
To learn the parameters $\mathbf{w}$ in a peer-to-peer fashion, we can consider a *random walk* procedure. At each step $t$ of the training process, a client $s$ receives the current model from one of its neighbors and updates this model with a gradient obtained using (a batch of) its private dataset. Subsequently, the client samples one of its neighbors $j \in \mathcal{N}(s)$, based on a predefined distribution over $\mathcal{N}(s)$, and forwards the updated model along with any other relevant optimization parameters. Algorithm 1 summarizes the above in pseudocode. Each client can run additional routines to evaluate the model, and if the target metrics have been achieved, stop the training and propagate the final model across the communication graph such

that every client has the most up-to-date copy. This could be based on aggregate validation metrics across clients, or some collaborative decision-making process, like majority voting.

**Designing the transition probability** In order to optimize Eq. 1, we have to construct a random walk procedure that has the correct stationary distribution over the clients, *i.e.*, $p(s) = N_s/N$. This will guarantee that, asymptotically, we are optimizing the global loss function. Let $\mathbf{A} \in \{0, 1\}^{S \times S}$ be an adjacency matrix that denotes the connectivity according to $\mathcal{G}$; if node $i$ is connected to $j$, then $\mathbf{A}_{ij} = \mathbf{A}_{ji} = 1$ and zero otherwise. Now consider $\mathbf{P}$ to be a simple $S \times S$ transition matrix between the $S$ nodes, such that the transition from a node $j$ to a node $i$ is given as $\mathbf{P}_{ij} = p(i|j) = 1/|\mathcal{N}(j)|, \forall i \in \mathcal{N}(j)$ and zero otherwise. $|\mathcal{N}(\cdot)|$ returns the amount of neighbors of a specific node (*i.e.*, its degree). In this case, we can define $\mathbf{D}$ as a diagonal matrix with the degrees of each of the nodes in the diagonal. Therefore, this simple strategy where each client samples a specific neighbor uniformly at random can be encoded as $\mathbf{P} = \mathbf{A}\mathbf{D}^{-1}$. This transition probability requires only local information and is therefore appropriate for the decentralized setting.

Unfortunately, this simple strategy does not generally have the desired stationary distribution. We can, however, impose a specific stationary distribution via a Metropolis-Hastings adjustment [3] of this simple proposal distribution:

$$\hat{\mathbf{P}}_{ij} = \begin{cases} \mathbf{P}_{ij} \min\left(1, \frac{p(i)\mathbf{P}_{ji}}{p(j)\mathbf{P}_{ij}}\right), & i \neq j \\ 1 - \sum_{k \neq j} \hat{\mathbf{P}}_{kj}, & i = j, \end{cases} \qquad \hat{\mathbf{P}}_{ij} = \begin{cases} \mathbf{P}_{ij} \min\left(1, \frac{N_i\mathbf{P}_{ji}}{N_j\mathbf{P}_{ij}}\right), & i \neq j \\ 1 - \sum_{k \neq j} \hat{\mathbf{P}}_{kj}, & i = j, \end{cases} \quad (2)$$

since the normalizing constant of the desired stationary distribution is the same in each node. This new transition probability guarantees that the stationary distribution is $p(s)$ and still requires only local information sharing (*i.e.*, data counts, and the number of neighbors), which can be done at the beginning of training.

**Structure of the communication graph** For a random walk procedure to be effective, we need a communication graph that facilitates fast "mixing." In other words, we need a communication graph $\mathcal{G}$ that allows the random walk to converge quickly to its stationary distribution. The main graph structure we assume in this work is that of "small-world" graphs [23]. Such graphs arise naturally in the context of social networks, wikis, and the internet. The main property of small-world graphs is that, while they are sparsely connected, the transition between any two pairs of nodes can be done in a small number of steps. Intuitively, they can be understood as ring graphs where some random connections have been replaced or added, with the new connections acting as "shortcuts" that allow for fast traversal.

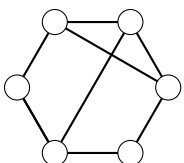

Figure 1: A small-world graph

## 4 Random Walk with Adaptive Optimization

Adaptive optimization methods, such as Adam [7], use exponential moving averages of gradients (first moment) and squared gradients (second moment) to correct for stochasticity in the gradient direction and adaptively increase or decrease the effective learning rate in response to the gradient variance across iterations. Each moment is equivalent in size to the gradient vector and needs to be stored between optimization steps. In centralized settings, the moments can be stored on the server and do not need to be transferred to the clients. In a decentralized setting, however, each client needs to have access to these moments, and thus, they need to be transmitted along with other updates.

### 4.1 Reducing communication costs

Transmitting the moments of the optimizer on each step of the random walk increases communication costs considerably. In this section, we propose two techniques that can effectively tackle this problem.

**Quantization** Our first strategy for reducing the communication cost is the compression of the state of Adam. While it generally is possible to compress both moments, we choose to drop the first moment (*i.e.*, set $\beta_1 = 0$), analogously to [18]. Not only does it help to save communication budget, but it also simplifies the convergence proof because compression of this moment would introduce bias in the update direction. Compression of the second moment can be done, for example, through scalar quantization or factorisation [18, 21].

**Algorithm 2** Decentralized learning by a random walk with compressed-state Adam and multiple local steps

> Client $s$ receives a message $\mathcal{M}$ from neighbor $h$
> $\mathbf{w}^{(t)}, \texttt{comp}(\mathbf{m}_2), t \leftarrow \mathcal{M}$
> $\mathbf{w}_1 \leftarrow \mathbf{w}^{(t)}$
> $\mathbf{m}_2 \leftarrow \texttt{decomp}(\texttt{comp}(\mathbf{m}_2))$
> **for** $k \leftarrow 1, \ldots, K-1$ **do**
> $\quad \mathbf{g}_{sk} \leftarrow \nabla_{\mathbf{w}} f(\mathbf{w}_k, \mathcal{D}_s)$
> $\quad \mathbf{m}_2 \leftarrow \beta_2 \mathbf{m}_2 + (1 - \beta_2)\mathbf{g}_{sk}^2$
> $\quad \hat{\mathbf{m}}_2 \leftarrow \mathbf{m}_2/(1 - \beta_2^{tK+k})$
> $\quad \mathbf{w}_{k+1} \leftarrow \mathbf{w}_k - \eta\mathbf{g}_{sk}/(\sqrt{\hat{\mathbf{m}}_2} + \epsilon)$
> **end for**
> $\mathbf{w}^{(t+1)} \leftarrow \mathbf{w}_K$
> $j \leftarrow$ random neighbor according to $\hat{\mathbf{P}}_{:s}$
> **return** Send $\mathbf{w}^{(t+1)}, \texttt{comp}(\mathbf{m}_2), t+1$ to neighbor $j$

**Algorithm 3** Neighbor selection procedure for a client $s$ according to $\hat{\mathbf{P}}_{:s}$

> $j \leftarrow$ random neighbor from $\mathcal{N}(s)$
> $C = \min\left(\frac{N_j|\mathcal{N}(s)|}{N_s|\mathcal{N}(j)|}, 1\right),$
> $u \sim \mathcal{U}[0, 1]$
> **if** $u < C$ **then**
> $\quad$ next_node $\leftarrow j$
> **else**
> $\quad$ next_node $\leftarrow s$
> **end if**
> **return** next_node

In this work, we introduce a simple, effective, and novel scalar quantization procedure for the second moment: Quantization in the log domain. More specifically, our procedure is motivated by the observation that most of the values of the second moment are quite small and that uniform quantization in the log domain provides more resolution for these lower values. Let $\mathbf{v}$ be the second moment: we use a single bit to denote whether $\mathbf{v}$ is non-zero and then use $b-1$ bits to do uniform quantization on $\log \mathbf{v}$ for the non-zero values on $\mathbf{v}$. We choose a uniform quantization strategy that represents the minimum and maximum values of the non-zero values of $\log \mathbf{v}$ exactly; thus, there is no clipping.

**Multiple local steps** Another strategy that can reduce the communication cost is to do multiple local steps on each client, popularized by FedAvg [15] in the server-orchestrated setting. While simple in nature, to the best of our knowledge, such a strategy has not been considered before in the random-walk type of optimization literature. We thus introduce the notion of multiple local updates in the random-walk type of optimization settings. More specifically, each client updates the model $K$ times on their local dataset before continuing the random walk. As a result, for a total of $M$ model updates, we only need to (roughly) communicate $M/K$ times, thus cutting down the overall communication cost by a factor of $K$.

Algorithm 2 presents the client-side logic of Adam with compressed moments and multiple local updates, whereas Algorithm 3 in presents the strategy for selecting the next node in the random walk.

### 4.2 Convergence analysis

In this section, we analyze the convergence behavior of decentralized-FL using random walks with no-momentum Adam and our proposed modifications in the non-convex setting. More specifically, we investigate the convergence rate in three settings: (1) random walk with Adam, (2) random walk with Adam and second-moment quantization, and (3) random walk with Adam, second-moment quantization, and multiple local updates. All the proofs are provided in Appendix A.

One of the main challenges for the theoretical analysis of this setting is in considering the bias in the gradient distribution that stems from performing a random walk to sample the gradients. In a similar fashion to [20], we provide the following lemma that characterizes this bias.

**Lemma 4.1.** *Let* $\mathbf{A} \in \{0, 1\}^{S \times S}$ *be an adjacency matrix of a connected, non-bipartite and undirected graph with $S$ nodes, and $\mathbf{P}$ a transition matrix (not necessarily symmetric) between the nodes that respects the connectivity of $\mathbf{A}$. Let the marginal distribution of the random walk over the nodes at timestep $t$ be*

$$\boldsymbol{\pi}_t = \mathbf{P}^t \boldsymbol{\pi_0}, \tag{3}$$

*where $\boldsymbol{\pi}_0$ is the initial distribution over the nodes. Let $\lambda_1, \ldots, \lambda_S$ be the eigenvalues of $\mathbf{P}$. Provided that $1 = \lambda_1 > \lambda_2 \geq \cdots \geq \lambda_S > -1$ we have that*

$$\left| \mathbb{E}_{\boldsymbol{\pi}_t}[f] - \mathbb{E}_{\boldsymbol{\pi}^*}[f] \right| \leq G\sqrt{N}\lambda^t, \tag{4}$$

*where $\boldsymbol{\pi}^*$ is the stationary distribution of the random walk, $|f|_\infty \leq G$ and $\lambda = \max(\lambda_2, |\lambda_N|)$.*

Beside this lemma, we also make the following standard assumptions in the non-convex optimization and federated learning literature [16].

**Assumption 1** (Lipschitz gradient). *Each local loss function $f_s$ is $L$-smooth $\forall s \in \mathcal{S}$, i.e., $\|\nabla f_s(\boldsymbol{x}) - \nabla f_s(\boldsymbol{y})\| \leq L\|\boldsymbol{x} - \boldsymbol{y}\|, \forall \boldsymbol{x}, \boldsymbol{y} \in \mathbb{R}^D$. The global loss function $f = \sum_s \frac{N_s}{N} f_s$ is also $L$-smooth.*

**Assumption 2** (Bounded gradient norm). *The stochastic gradient norm is upper bounded by a constant $G$*

$$\|\nabla f(\mathbf{w}, \xi)\|_\infty \leq G, \quad \forall \xi \in \mathcal{D}_s, \forall s, \forall \mathbf{w} \in \mathbb{R}^D. \tag{5}$$

**Assumption 3** (Bounded gradient variance). *For any dimension $i$, the gradient variance at a client $s$ is upper bounded by $\sigma_{li}^2$, $\forall s$, and the global gradient variance by $\sigma_{gi}^2$*

$$\mathbb{E}_{s,\xi}[(\nabla f(\mathbf{w}, \xi)_i - \nabla f(\mathbf{w})_i)^2] = \mathbb{E}_{s,\xi}\left[(\nabla f(\mathbf{w}, \xi)_i - \nabla f_s(\mathbf{w}))_i^2\right] + \mathbb{E}_s\left[(\nabla f_s(\mathbf{w})_i - \nabla f(\mathbf{w})_i)^2\right]$$

$$\leq \sigma_{li}^2 + \sigma_{gi}^2 = \sigma_i^2. \tag{6}$$

*Furthermore, we define $\sigma^2 = \sum_i \sigma_i^2$ and $\sigma_l^2 = \sum_i \sigma_{li}^2$.*

Based on the above, we can prove the following theorem about the convergence of random-walk optimization with Adam and no momentum.

**Theorem 4.1.** *Let $\mathbf{w} \in \mathbb{R}^D$ be the model parameter vector that is optimized for $T$ random walk steps. Under assumptions 1, 2, 3 and if the learning rate $\eta$ and $\epsilon$ are chosen such that $\eta \leq \frac{\epsilon}{2L}$ and $1 - \beta_2 \leq \frac{\epsilon^2}{16G^2}$ then we have that the updates of random walk Adam without momentum satisfy*

$$\mathbb{E}[\|\nabla f(\mathbf{w}_a)\|^2] \leq O\left(\frac{f(\mathbf{w}_1) - f(\mathbf{w}^*)}{\eta T} + \sigma^2 + \frac{\lambda D\sqrt{N}}{(1-\lambda)T}\right) \tag{7}$$

*where $\mathbf{w}_a$ is a randomly chosen iterate from $\mathbf{w}_1, \ldots, \mathbf{w}_T$.*

Therefore, we can see that the asymptotic bound we obtain is similar to the traditional Adam [25] with an additional error term stemming from decentralization which, however, decreases with the number of iterations.

We now consider the case of performing $b$-bit scalar quantization to the second moment of Adam. To the best of our knowledge, while a similar setting has been considered in practice by [5], it has not been theoretically analyzed before. In order to analyze its convergence, we need one more assumption for the quantization procedure.

**Assumption 4** (Bounded quantization noise). *Let $\mathbf{v}$ be a variable to be quantized and $Q(\cdot)$ the quantization operation. We assume that there is a constant $q \geq 0$ such that the quantization noise is bounded as*

$$\|Q(\mathbf{v}) - \mathbf{v}\|_1 \leq q\|\mathbf{v}\|_1.$$

Notice that our specific log-uniform quantization strategy conforms to this assumption since the upper bound on the quantization noise is given by half the step size of the uniform quantizer $\Delta$. For any given dimension $i$ we thus have $|Q(\mathbf{v}_i) - \mathbf{v}_i| \leq |\mathbf{v}_i \exp\left(\frac{\Delta}{2}\right) - \mathbf{v}_i| = |\exp\left(\frac{\Delta}{2}\right) - 1||\mathbf{v}_i|$. In this case, we can prove the following theorem.

**Theorem 4.2.** *Let $\mathbf{w} \in \mathbb{R}^D$ be the model parameter vector that is optimized for $T$ random walk steps. Under assumptions 1, 2, 3, 4 and if the learning rate $\eta$ and $\epsilon$ are chosen such that $\eta \leq \frac{\epsilon}{2L}$ and $1 - \beta_2 \leq \frac{\epsilon^2}{16G^2}$ then we have that the updates of random walk Adam without momentum and with second moment quantization satisfy*

$$\mathbb{E}[\|\nabla f(\mathbf{w}_a)\|^2] \leq \mathcal{O}\left(\frac{f(\mathbf{w}_1) - f(\mathbf{w}^*)}{\eta T} + \sigma^2 + qDG + \frac{\lambda D\sqrt{N}}{(1-\lambda)T}\right) \tag{8}$$

*where $\mathbf{w}_a$ is a randomly chosen iterate from $\mathbf{w}_1, \ldots, \mathbf{w}_T$.*

For the second setting, we consider the case where each client does a fixed number of updates $K$ before deciding whether to send the model to one of its neighbors. While such a case has been analyzed theoretically before in gossip type of approaches [10], it has not been analyzed for random walks. In this specific case, we require one more assumption, which we can use to prove Theorem 4.3.

**Assumption 5** (Bounded difference between local and global gradients). *For any given point* $\mathbf{w} \in \mathbb{R}^D$ *there is a constant* $\zeta^2$ *that upper bounds the squared* $L_2$ *distance between the gradient on any given client* $s$ *and the global gradient:*

$$\|\nabla f_s(\mathbf{w}) - \nabla f(\mathbf{w})\|_2^2 \leq \zeta^2. \tag{9}$$

**Theorem 4.3.** *Let* $\mathbf{w} \in \mathbb{R}^D$ *be the model parameter vector that is optimized for* $T$ *random walk steps and let* $K$ *be the number of updates a client does before continuing the random walk. Under assumptions 1, 2, 3, 4, 5 and if the learning rate* $\eta$ *and* $\epsilon$ *are chosen such that* $\eta \leq \frac{\epsilon}{4L}$ *and* $1 - \beta_2 \leq \frac{\epsilon^2}{64G^2}$ *then we have that the updates of random walk Adam without momentum and second moment quantization satisfy*

$$\mathbb{E}[\|\nabla f(\mathbf{w}_a)\|^2] \leq \mathcal{O}\left( \frac{f(\mathbf{w}_1) - f(\mathbf{w}^*)}{\eta KT} + \frac{\sigma^2}{K} + qDG + \frac{K-1}{K}(\sigma_l^2 + \zeta^2 + G^2) + \frac{\lambda D \sqrt{N}}{(1-\lambda)KT} \right) \tag{10}$$

*where* $\mathbf{w}_a$ *is a randomly chosen iterate from* $\mathbf{w}_1, \ldots, \mathbf{w}_{KT}$.

The bound is intuitive. We can see that we are essentially trading total variance $\sigma^2$ for local gradient variance $\sigma_l^2$ and bias $\zeta$. As a result, this strategy will be useful whenever the data are close to i.i.d. across the clients; in this case, $\zeta^2$ will be small, and since $\sigma_l^2 \leq \sigma^2$, it will be beneficial to do multiple local steps $K$ to reduce the number of communication rounds $T$ required.

## 5 Evaluation

We consider several established centralized federated learning benchmarks (vision and text) to evaluate the performance of our decentralized random walk optimization procedures with a variety of models. Vision tasks include training ResNet-20 on non-i.i.d. splits of CIFAR-10, CIFAR-100, with 100 and 500 clients respectively, as described in [16]; LeNet-5 model on the naturally non-i.i.d. split of FEMNIST (partitioned by writer IDs into $\sim$3.6k clients) [13]. Language tasks consist of next word prediction using an LSTM model on a non-i.i.d. split of the Shakespeare dataset into 660 clients [15], and the StackOverflow tag prediction with a logistic regression model [16] and $\sim$ 342k clients.

For the majority of experiments, we used a "small-world" graph with an average degree of five, implemented using the Watts–Strogatz model [23], with $\beta = 0.5$. Unless mentioned otherwise, we use 4-bit per-tensor quantization for the second moment, and we consider a separate quantization grid for the weights of each layer. Empirically, we found it beneficial to employ stochastic (in place of deterministic) rounding in log-space for quantization. In the case of multiple local steps, we only quantize the second moment whenever it is transmitted and not in-between the local optimization steps. More details about the experimental setup can be found in the Appendix.

### 5.1 Comparison with other FL methods

**Comparison against centralized FL**   First, we evaluate the performance of random walk learning against a centralized baseline (FedAvg with Adam on the server [16] and one epoch of updates on each client). Figure 2 depicts validation accuracy as a function of total communication costs (both upload and download) for FedAvg and different versions of random walk optimization. As the StackOverflow task consists of large validation and test sets, we plot the accuracy on the first 10% of validation examples. The final numbers (including evaluation on the full test set) are provided in Table 1.

We observe that even a simple RW-SGD performs well on vision tasks, provided that the learning rate is chosen appropriately. Its accuracy on CIFAR-10 and FEMNIST is on par with or better than FedAvg, while communication is lower (in case of FEMNIST, by 32%). RW-Adam provides competitive performance even without hyperparameter tuning, thanks to learning rate adaptation via

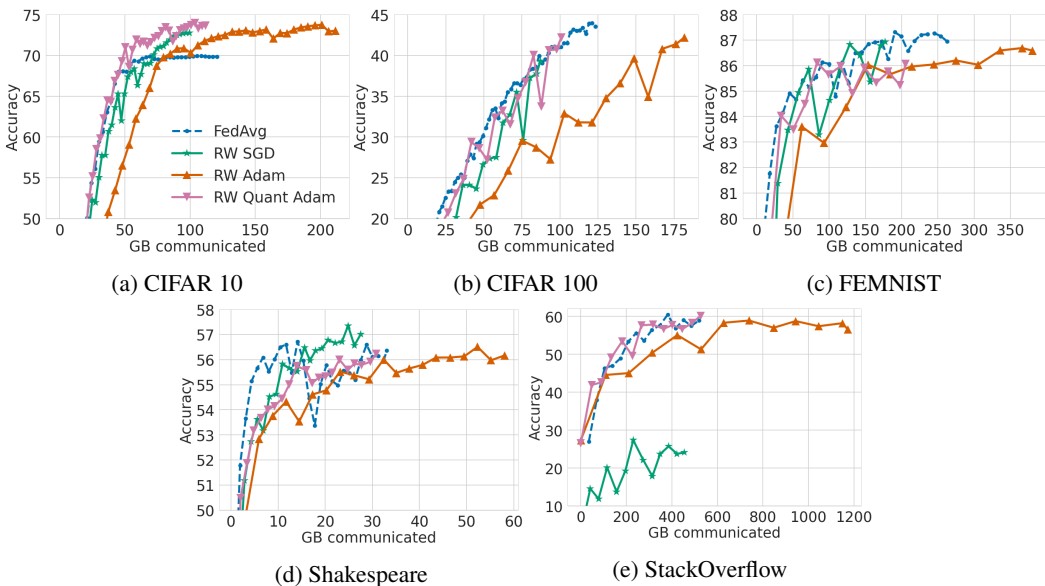

Figure 2: Accuracy as a function of the cumulative communication costs on the tasks considered.

Table 1: Test-set accuracy at the end of training and total communication (in GB).

| | CIFAR10 $K=1, b=4$ | | CIFAR100 $K=1, b=5$ | | FEMNIST $K=1, b=5$ | | Shakespeare $K=10, b=4$ | | StackOverflow $K=1, b=5$ | |
| Method | Acc. | Comm. | Acc. | Comm. | Acc. | Comm. | Acc. | Comm. | Acc. | Comm. |
|---|---|---|---|---|---|---|---|---|---|---|
| FedAvg | 69.9 | 120 | 43.4 | 121 | 86.9 | 263 | 56.4 | 33 | 61.4 | 522 |
| RW SGD | 72.8 | 99 | 39.3 | 86 | 87.0 | 177 | 57.0 | 28 | 23.5 | 463 |
| RW Adam | 73.1 | 211 | 42.2 | 180 | 86.6 | 302 | 56.2 | 58 | 62.5 | 1192 |
| RW QAdam | 73.7 | 112 | 42.2 | 100 | 86.3 | 163 | 56.2 | 31 | 62.1 | 535 |

the second moment. Our quantization procedure brings communication costs down to the level of RW-SGD without impact on accuracy.

When gradients are sparse, like in the case of StackOverflow, or the task is more complex (CIFAR-100), we see that adaptive optimization is crucial to obtain a better model. On CIFAR-100, RW-Adam provides a 7% boost in validation accuracy compared to RW-SGD and gets closer to FedAvg. Even more noticeably, on StackOverflow, RW-Adam maintains near-FedAvg performance, while RW-SGD only reaches less than a half of centralized performance within a comparable communication budget.

Finally, we observe that Adam with the default hyperparameters shows robust performance in all the experiments involving a neural network. At the same time, for SGD, we find it necessary to tune parameters on a per-task basis. For instance, vision tasks run with a learning rate of 0.1, while 1.0 is used for Shakespeare.

**Comparison with gossiping** To compare gossiping with random walks, we run both algorithms on CIFAR-10 and the Shakespeare dataset. In both cases, we use the same small-world setup as before. The choice of datasets is partially dictated by the limitations of gossiping for large-scale federations since the other tasks have a significantly larger number of clients. In Table 2, we can observe that the raw gossip averaging communication far exceeds both FedAvg and random walks. For instance, it requires more than 2TB on CIFAR-10 to reach over 60% accuracy, which is significantly lower than what is achieved by other methods. Even, hypothetically, applying PowerGossip compression [21] to the updates (which would further reduce validation accuracy), the overall communication remains high ($\sim$ 41GB), and RW-QAdam reaches better performance. Similarly, for Shakespeare: over 400GB of raw communication is necessary for 40% accuracy, and

Table 2: Comparison of decentralized paradigms. Accuracy and total communication (in GB; for gossiping, marked with *, accounts for PowerGossip compression, full communication in parenthesis).

| | CIFAR10 | | Shakespeare | |
| Method | Acc. | Comm. | Acc. | Comm. |
|---|---|---|---|---|
| Gossip Averaging | 63.5 | 41 (2328)* | 45.3 | 4.4 (442.7)* |
| RW QAdam | 72.6 | 41 | 51.9 | 3.4 |

even when accounting for possible compression, gossiping performance is considerably worse than RW-QAdam with less communication.

## 5.2 Additional studies

**Impact of the communication graph**    To empirically measure the dependence of our method on the communication graph topology, we conduct a study on CIFAR-10 with 100 clients. We fix RW-QAdam with 5 bits for the second moment and consider three specific communication graphs: (a) a fully connected graph, (b) a small-world graph with an average degree of 5 (*i.e.*, as in all other experiments) and (c) a ring graph with a degree of 2. Figure 3a shows the convergence speed of random walk training. We see that the small world and the fully connected graph have similar convergence (indicating good "mixing"), whereas the ring topology is lagging behind.

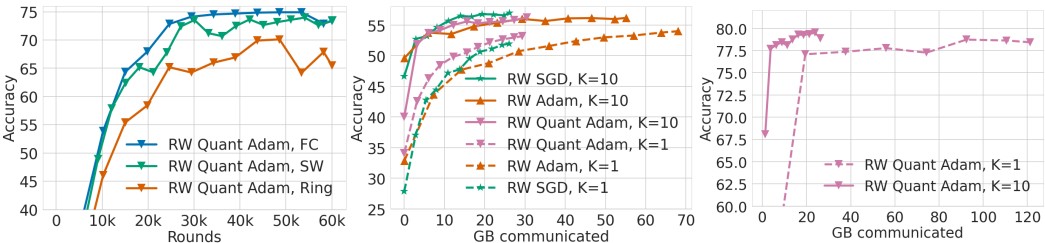

(a) Training with different graphs.    (b) Shakespeare with different $K$.    (c) i.i.d. CIFAR 10 with different $K$.

Figure 3: Additional studies. (a) Performance on CIFAR-10 with different communication graph topologies. (b, c) Effect of multiple local updates on Shakespeare and a CIFAR-10 i.i.d. split.

**Impact of multiple local updates**    We argue for doing multiple local updates to improve performance while reducing communication costs. Indeed, our theory suggests that if the data are close to i.i.d. across the nodes, multiple local updates can be beneficial. Empirically, we observe that $K = 1$ performed better on many (artificially or naturally) non-i.i.d. tasks, except Shakespeare. In Figure 3b, we show that on the Shakespeare dataset, there is a big improvement in the communication vs. accuracy tradeoff when we move from $K = 1$ to $K = 10$ local updates on each client. To further test our theory, we set up an i.i.d. CIFAR-10 task with 100 clients. Figure 3c depicts the communication vs accuracy tradeoffs of our quantized state Adam with $K = 1$ and $K = 10$ local updates. We can similarly see the benefits of multiple local steps as our theory predicted. As a reference, non-federated training reaches a test set accuracy of $78.9\%$ in this setting, which is close to the performance of our random walks.

## 6  Conclusion

In this paper, we proposed a random-walk-based learning algorithm with compressed-state adaptive optimization to solve the problem of federated learning without a central server in a peer-to-peer fashion. Since the communication graph of participating devices can be formed based on proximity or connection speeds, this approach has the potential to minimize the overall time spent on communication. Despite the increase in the number of rounds necessary for convergence, the total communication cost of our approach is on par or better than that of federated averaging because each round involves communication between only two clients (as opposed to typically 10–100 in

FedAvg). Our evaluation showed that compressed adaptive optimization on top of a random walk enables performance comparable with centralized FL solutions, even on sparsely connected graphs.

One potential disadvantage of random walk learning is that client updates are sequential, while in FedAvg or gossiping, they can be done in parallel. However, this drawback can be compensated by parallelizing hyperparameter search and running multiple random walks with different configurations. Since there is no dependency between them, no synchronization is needed in this process, allowing to bypass stragglers and effectively gauge the performance of multiple models. We leave a detailed methodology and evaluation of this approach for future work. Another important aspect for future study is combining our approach with formal privacy notions. For example, [4] have proposed network differential privacy and showed that it is well suited for simple random walks; continuing this line of research for adaptive and communication-efficient random walk optimizers is an interesting future direction.

Conceptually, a decentralized FL approach removes reliance on a central server, the independence and trustworthiness of which cannot be taken for granted in many parts of the world. Beyond moving the control over the data to the user through FL, our work takes a step towards putting users in control over the entire learning process itself. The immediate societal consequence of such an approach is more resilience to central interference but also less accountability and a higher potential of misuse by individuals. Developing robust mechanisms against diverse forms of attacks will be crucial to establishing decentralized FL in the future.

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

# Appendix

## A Convergence proof in the case of a random walk

### A.1 Proof of Lemma 4.1

Let $\mathbf{A}$ be an undirected adjacency matrix that denotes the connectivity between $N$ nodes; if node $i$ is connected to $j$, then $\mathbf{A}_{ij} = \mathbf{A}_{ji} = 1$ and zero otherwise. Now also let $\mathbf{P}$ be the $N \times N$ transition matrix between the $N$ nodes, where we have that a transition from a node $j$ to a node $i$ is given as $\mathbf{P}_{ij} = p(i|j) = 1/d(j) \ \forall i \in \mathcal{N}(j)$ and zero otherwise, where $d(\cdot)$ is a function that returns the amount of neighbors of a specific node (*i.e.*, the degree). In this case, we have that $\mathbf{P} = \mathbf{A}\mathbf{D}^{-1}$ where $\mathbf{D}$ is a diagonal matrix having the degrees of each of the nodes in the diagonal. Having access to the above, we can compute the marginal distribution of being in a specific node $i$ at time $t$ as $\pi_t(i) = \sum_j \pi_{t-1}(j)p(i|j)$. This can be compactly written as

$$\boldsymbol{\pi}_t = \mathbf{A}\mathbf{D}^{-1}\boldsymbol{\pi}_{t-1} = (\mathbf{A}\mathbf{D}^{-1})^t\boldsymbol{\pi}_0, \tag{11}$$

where $\boldsymbol{\pi}_0$ is the initial distribution over the nodes (*e.g.*, a one-hot vector when we deterministically start the random walk from a specific location). The stationary distribution of the walk will be $\boldsymbol{\pi}^*$ and it will satisfy the relation $\boldsymbol{\pi}^* = \mathbf{A}\mathbf{D}^{-1}\boldsymbol{\pi}^*$.

For the purposes of this work, we are interested in bounding the divergence of $\boldsymbol{\pi}^t$ from the stationary distribution $\boldsymbol{\pi}^*$, as in our case $\boldsymbol{\pi}^*$ corresponds to sampling the global gradient distribution. To do that, we will follow the analysis described at [34]. Let $\mathbf{C} = \mathbf{D}^{-1/2}\mathbf{A}\mathbf{D}^{-1/2}$ be the *normalized* connected and non-bipartite adjacency matrix which will have that its eigenvalues will be $1 = \lambda_1 > \lambda_2 \geq \cdots \geq \lambda_N > -1$ and its corresponding orthonormal eigenvectors will be $\mathbf{b}_1, \ldots, \mathbf{b}_N$. As $\mathbf{A}\mathbf{D}^{-1}$ is *similar* to $\mathbf{D}^{-1/2}\mathbf{A}\mathbf{D}^{-1}\mathbf{D}^{1/2} = \mathbf{D}^{-1/2}\mathbf{A}\mathbf{D}^{-1/2} = \mathbf{C}$, we will have that $\mathbf{A}\mathbf{D}^{-1}$ has the same eigenvalues as $\mathbf{C}$. Furthermore, we will have that its eigenvectors will be

$$\mathbf{D}^{-1/2}\mathbf{A}\mathbf{D}^{-1}\mathbf{D}^{1/2}\mathbf{b}_i = \mathbf{C}\mathbf{b}_i = \lambda_i\mathbf{b}_i \quad \rightarrow \quad \mathbf{A}\mathbf{D}^{-1}\mathbf{D}^{1/2}\mathbf{b}_i = \lambda_i\mathbf{D}^{1/2}\mathbf{b}_i, \tag{12}$$

thus the eigenvector for the $i$'th eigenvalue of $\mathbf{A}\mathbf{D}^{-1}$ will be $\hat{\mathbf{b}}_i = \mathbf{D}^{1/2}\mathbf{b}_i$. Let the matrix of eigenvectors be denoted as $\hat{\mathbf{B}} = \mathbf{D}^{1/2}\mathbf{B}$, where $\mathbf{B}$ is the set of eigenvectors for $\mathbf{C}$. Now, since the set of eigenvectors constitute a basis, we can express the initial distribution $\boldsymbol{\pi}_0$ under this basis, $\boldsymbol{\pi}_0 = \sum_{i=1}^N c_i\hat{\mathbf{b}}_i$, where $c_i = \hat{\mathbf{b}}_i^T\boldsymbol{\pi}_0$. We will then have that

$$\boldsymbol{\pi}_t = (\mathbf{A}\mathbf{D}^{-1})^t\boldsymbol{\pi}_0 = (\hat{\mathbf{B}}\boldsymbol{\Lambda}\hat{\mathbf{B}}^T)^t\boldsymbol{\pi}_0 = \hat{\mathbf{B}}\boldsymbol{\Lambda}^t\hat{\mathbf{B}}^T\sum_{i=1}^N c_i\hat{\mathbf{b}}_i = \sum_{i=1}^N c_i\lambda_i^t\hat{\mathbf{b}}_i. \tag{13}$$

Now by taking the limit of $t \rightarrow \infty$ we can see that the stationary distribution of the above is

$$\boldsymbol{\pi}^* = \lim_{t\to\infty}\sum_{i=1}^N c_i\lambda_i^t\hat{\mathbf{b}}_i = \sum_{i=1}^N \lim_{t\to\infty} c_i\lambda_i^t\hat{\mathbf{b}}_i = c_1\lambda_1\hat{\mathbf{b}}_i = c_1\hat{\mathbf{b}}_1, \tag{14}$$

as $\lambda_1 = 1$ and is the only eigenvalue that doesn't decay with $t$. Therefore, we have that the stationary distribution will be associated with the first eigenvector of $\mathbf{A}\mathbf{D}^{-1}$. We can now return to our original question; finding a bound between the distribution of the chain at timestep $t$ and the stationary distribution. We will consider the total variation distance, *i.e.*,

$$TV(\boldsymbol{\pi}_t, \boldsymbol{\pi}^*) = \frac{1}{2}\|\boldsymbol{\pi}_t - \boldsymbol{\pi}^*\|_1 = \frac{1}{2}\left\|(\mathbf{A}\mathbf{D}^{-1})^t\boldsymbol{\pi}_0 - \boldsymbol{\pi}^*\right\|_1 \tag{15}$$

$$= \frac{1}{2}\left\|\sum_{i=1}^N c_i\lambda_i^t\hat{\mathbf{b}}_i - \boldsymbol{\pi}^*\right\|_1 = \frac{1}{2}\left\|\boldsymbol{\pi}^* + \sum_{i=2}^N c_i\lambda_i^t\hat{\mathbf{b}}_i - \boldsymbol{\pi}^*\right\|_1 \tag{16}$$

$$= \frac{1}{2}\left\|\sum_{i=2}^N c_i\lambda_i^t\hat{\mathbf{b}}_i\right\|_1 \tag{17}$$

$$\leq \frac{\sqrt{N}}{2}\left\|\sum_{i=2}^N c_i\lambda_i^t\hat{\mathbf{b}}_i\right\|_2. \tag{18}$$

We also have that

$$\left\| \sum_{i=2}^{N} c_i \lambda_i^t \hat{\mathbf{b}}_i \right\|_2^2 = \sum_{i=2}^{N} c_i^2 \lambda_i^{2t}, \tag{19}$$

due to the $\hat{\mathbf{b}}_i$ forming an orthonormal basis. By taking $\lambda = \max(\lambda_2, |\lambda_N|)$, we can now bound the previous in the following way

$$\sum_{i=2}^{N} c_i^2 \lambda_i^{2t} \le \lambda^{2t} \sum_{i=2}^{N} c_i^2 \le \lambda^{2t} \sum_{i=1}^{N} c_i^2 \le \lambda^{2t}, \tag{20}$$

where the last inequality is due to

$$1 \ge \|\boldsymbol{\pi}_0\|_2^2 = \left\| \sum_{i=1}^{N} c_i \hat{\mathbf{b}}_i \right\|_2^2 = \sum_{i=1}^{N} c_i^2. \tag{21}$$

Therefore, we have that

$$\left\| \sum_{i=2}^{N} c_i \lambda_i^t \hat{\mathbf{b}}_i \right\|_2^2 \le \lambda^{2t} \qquad \rightarrow \qquad \left\| \sum_{i=2}^{N} c_i \lambda_i^t \hat{\mathbf{b}}_i \right\|_2 \le \lambda^t \tag{22}$$

and thus

$$TV(\boldsymbol{\pi}_t, \boldsymbol{\pi}^*) \le \frac{\sqrt{N}}{2} \lambda^t. \tag{23}$$

Now, from the alternative representation of the total variation distance, we have that for $f$ in a space of functions $F$ with an image in $\mathbb{R}$ that takes as inputs samples from distributions $p, q$, $TV(p, q) = \frac{1}{2} \sup_{f \in F, |f| \le 1} (\mathbb{E}_p[f] - \mathbb{E}_q[f])$ [29]. This can be generalized to arbitrary functions $f$ bounded by $G$; in this case, we have that $TV(p, q) = \frac{1}{2G} \sup_{f \in F, |f| \le G} (\mathbb{E}_p[f] - \mathbb{E}_q[f])$. Therefore, applying this result to our distributions over selecting the nodes, we have that

$$TV(\boldsymbol{\pi}_t, \boldsymbol{\pi}^*) = \frac{1}{2G} \sup_{f \in F, |f| \le G} \left( \mathbb{E}_{\boldsymbol{\pi}_t}[f] - \mathbb{E}_{\boldsymbol{\pi}^*}[f] \right) \le \frac{\sqrt{N}}{2} \lambda^t \tag{24}$$

and, thus, for any $f \in F$ with $|f| \le G$ we have that

$$\mathbb{E}_{\boldsymbol{\pi}_t}[f] \le \mathbb{E}_{\boldsymbol{\pi}^*}[f] + G\sqrt{N} \lambda^t. \tag{25}$$

We can see that this bound has a sublinear dependence on the number of nodes $N$. Furthermore, as the total-variation distance is symmetric, we further have that

$$\mathbb{E}_{\boldsymbol{\pi}^*}[f] \le \mathbb{E}_{\boldsymbol{\pi}_t}[f] + G\sqrt{N} \lambda^t. \tag{26}$$

Therefore, both of these facts can be combined to yield

$$\left| \mathbb{E}_{\boldsymbol{\pi}_t}[f] - \mathbb{E}_{\boldsymbol{\pi}^*}[f] \right| \le G\sqrt{N} \lambda^t, \tag{27}$$

which constitutes Lemma 4.1.

To analyze the transition matrix arising from the Metropolis-Hastings adjustment of the simple proposal, $\hat{\mathbf{P}}$, we can work with the similar matrix

$$\mathbf{S} = \mathbf{R}^{1/2} \hat{\mathbf{P}} \mathbf{R}^{-1/2}, \tag{28}$$

here $\mathbf{R}$ is a diagonal matrix that has the desired stationary distribution, $\boldsymbol{\pi}^*$, in its diagonal. From [31] we know that the matrix $\mathbf{S}$ is symmetric (thus has only real eigenvalues) and satisfies the property that $1 = \lambda_1 > \lambda_2 \cdots \ge \lambda_N > -1$. Now let the eigenvectors of $\mathbf{S}$ be $\mathbf{s}_i$, we can show that the left eigenvectors of $\hat{\mathbf{P}}$ are $\mathbf{s}_i^T \mathbf{R}^{1/2}$ and the right eigenvectors are $\mathbf{R}^{-1/2} \mathbf{s}_i$. From [31] we also know that the left and right eigenvectors of $\hat{\mathbf{P}}$ associated with $\lambda_1 = 1$ are $\boldsymbol{\pi}^*$ and $\mathbf{1}$ respectively, therefore $c_1 \hat{\mathbf{b}}_1 = \boldsymbol{\pi}^*$. As a result, the same analysis and bounds apply.

## A.2 Convergence analysis in the non-convex case

We will base this proof on the proof technique of [25] for the case of Adam with only the second moment; the main idea is to use the previous bounds in order to translate the expectation over the Markov chain to an expectation over the stationary gradient distribution. With $\mathbf{w}_t$ we will denote the vector of optimized parameters at timestep $t$, with $\mathbf{v}_t$ the vector of the second moments per parameter at timestep $t$ and with $\mathbf{g}_t$ the stochastic gradient obtained at timestep $t$. we will We start with the update rule of Adam (without the first moment) for a single dimension $i$

$$\mathbf{w}_{t+1,i} = \mathbf{w}_{t,i} - \eta_t \frac{\mathbf{g}_{t,i}}{\sqrt{\mathbf{v}_{t,i}} + \epsilon}, \qquad \mathbf{w}_{t+1,i} - \mathbf{w}_{t,i} = -\eta_t \frac{\mathbf{g}_{t,i}}{\sqrt{\mathbf{v}_{t,i}} + \epsilon}. \tag{29}$$

where both $\epsilon$ and the second moment $\mathbf{v}$ are non-negative. We then assume an L-Lipschitz loss function and proceed by:

$$f(\mathbf{w}_{t+1}) \leq f(\mathbf{w}_t) + \nabla f(\mathbf{w}_t)^T(\mathbf{w}_{t+1} - \mathbf{w}_t) + \frac{L}{2}\|\mathbf{w}_{t+1} - \mathbf{w}_t\|^2 \tag{30}$$

$$= f(\mathbf{w}_t) - \eta_t \sum_{i=1}^{D} \nabla f(\mathbf{w}_t)_i \frac{\mathbf{g}_{t,i}}{\sqrt{\mathbf{v}_{t,i}} + \epsilon} + \frac{L\eta_t^2}{2} \sum_{i=1}^{D} \frac{\mathbf{g}_{t,i}^2}{\left(\sqrt{\mathbf{v}_{t,i}} + \epsilon\right)^2}. \tag{31}$$

We then take an expectation over the distribution of the chain at timestep $t$, $\boldsymbol{\pi}_t$, along with the data samples $\xi$ given that we are at $\mathbf{w}_t$

$$\mathbb{E}_{\boldsymbol{\pi}_t, \xi}[f(\mathbf{w}_{t+1})] \leq f(\mathbf{w}_t) - \eta_t \sum_{i=1}^{D} \nabla f(\mathbf{w}_t)_i \mathbb{E}_{\boldsymbol{\pi}_t, \xi}\left[\frac{\mathbf{g}_{t,i}}{\sqrt{\mathbf{v}_{t,i}} + \epsilon}\right] + \frac{L\eta_t^2}{2} \sum_{i=1}^{D} \mathbb{E}_{\boldsymbol{\pi}_t, \xi}\left[\frac{\mathbf{g}_{t,i}^2}{\left(\sqrt{\mathbf{v}_{t,i}} + \epsilon\right)^2}\right], \tag{32}$$

and we then apply Lemma 4.1 to express the expectations in the r.h.s. to ones over the stationary distribution at timestep $t$, $\boldsymbol{\pi}^*$. For that, we need an upper bound on the functions, *i.e.*, $|f|_\infty \leq C$, that we take the expectation over. In this specific case, we have that

$$\left|\frac{\mathbf{g}_{t,i}}{\sqrt{\mathbf{v}_{t,i}} + \epsilon}\right| \leq \frac{|\mathbf{g}_{t,i}|}{\sqrt{\beta_2 \mathbf{v}_{t-1,i} + (1-\beta_2)\mathbf{g}_{t,i}^2}} \leq \frac{1}{\sqrt{1-\beta_2}}, \tag{33}$$

$$\frac{\mathbf{g}_{t,i}^2}{(\sqrt{\mathbf{v}_{t,i}} + \epsilon)^2} = \left|\frac{\mathbf{g}_{t,i}}{\sqrt{\mathbf{v}_{t,i}} + \epsilon}\right|\left|\frac{\mathbf{g}_{t,i}}{\sqrt{\mathbf{v}_{t,i}} + \epsilon}\right| \leq \frac{1}{1-\beta_2}, \tag{34}$$

and thus

$$\mathbb{E}_{\boldsymbol{\pi}_t, \xi}[f(\mathbf{w}_{t+1})] \leq f(\mathbf{w}_t) - \eta_t \sum_{i=1}^{D} \nabla f(\mathbf{w}_t)_i \mathbb{E}_{\boldsymbol{\pi}^*, \xi}\left[\frac{\mathbf{g}_{t,i}}{\sqrt{\mathbf{v}_{t,i}} + \epsilon}\right] + \eta_t \sum_{i=1}^{D} \nabla f(\mathbf{w}_t)_i \frac{\sqrt{N}\lambda^t}{\sqrt{1-\beta_2}} +$$

$$+ \underbrace{\frac{L\eta_t^2}{2} \sum_{i=1}^{D} \mathbb{E}_{\boldsymbol{\pi}^*, \xi}\left[\frac{\mathbf{g}_{t,i}^2}{\left(\sqrt{\mathbf{v}_{t,i}} + \epsilon\right)^2}\right] + \frac{L\eta_t^2 D\sqrt{N}\lambda^t}{2(1-\beta_2)}}_{T_{2t}} \tag{35}$$

$$\leq f(\mathbf{w}_t) - \eta_t \sum_{i=1}^{D} \nabla f(\mathbf{w}_t)_i \mathbb{E}_{\boldsymbol{\pi}^*, \xi}\left[\frac{\mathbf{g}_{t,i}}{\sqrt{\mathbf{v}_{t,i}} + \epsilon}\right] + \frac{\eta_t DG\sqrt{N}\lambda^t}{\sqrt{1-\beta_2}} +$$

$$+ \underbrace{\frac{L\eta_t^2}{2} \sum_{i=1}^{D} \mathbb{E}_{\boldsymbol{\pi}^*, \xi}\left[\frac{\mathbf{g}_{t,i}^2}{\left(\sqrt{\mathbf{v}_{t,i}} + \epsilon\right)^2}\right] + \frac{L\eta_t^2 D\sqrt{N}\lambda^t}{2(1-\beta_2)}}_{T_{2t}}. \tag{36}$$

We then proceed in a similar manner to [25]

$$\leq f(\mathbf{w}_t) - \eta_t \sum_{i=1}^{D} \nabla f(\mathbf{w}_t)_i \mathbb{E}_{\boldsymbol{\pi}^*, \xi}\left[\frac{\mathbf{g}_{t,i}}{\sqrt{\mathbf{v}_{t,i}} + \epsilon} + \frac{\mathbf{g}_{t,i}}{\sqrt{\beta_2 \mathbf{v}_{t-1,i}} + \epsilon} - \frac{\mathbf{g}_{t,i}}{\sqrt{\beta_2 \mathbf{v}_{t-1,i}} + \epsilon}\right] +$$

$$+ \frac{\eta_t DG\sqrt{N}\lambda^t}{\sqrt{1-\beta_2}} + T_{2t}. \tag{37}$$

Given that $\nabla f(\mathbf{w}_t) = \mathbb{E}_{\boldsymbol{\pi}^*,\xi}[\mathbf{g}_t]$, we can continue with

$$= f(\mathbf{w}_t) - \eta_t \sum_{i=1}^{D} \nabla f(\mathbf{w}_t)_i \left( \frac{\nabla f(\mathbf{w}_t)_i}{\sqrt{\beta_2 \mathbf{v}_{t-1,i}} + \epsilon} + \mathbb{E}_{\boldsymbol{\pi}^*,\xi} \left[ \frac{\mathbf{g}_{t,i}}{\sqrt{\mathbf{v}_{t,i}} + \epsilon} - \frac{\mathbf{g}_{t,i}}{\sqrt{\beta_2 \mathbf{v}_{t-1,i}} + \epsilon} \right] \right) +$$
$$+ \frac{\eta_t D G \sqrt{N} \lambda^t}{\sqrt{1 - \beta_2}} + T_{2t} \tag{38}$$

$$= f(\mathbf{w}_t) - \eta_t \sum_{i=1}^{D} \frac{\nabla f(\mathbf{w}_t)_i^2}{\sqrt{\beta_2 \mathbf{v}_{t-1,i}} + \epsilon} - \eta_t \sum_{i=1}^{D} \nabla f(\mathbf{w}_t)_i \mathbb{E}_{\boldsymbol{\pi}^*,\xi} \left[ \frac{\mathbf{g}_{t,i}}{\sqrt{\mathbf{v}_{t,i}} + \epsilon} - \frac{\mathbf{g}_{t,i}}{\sqrt{\beta_2 \mathbf{v}_{t-1,i}} + \epsilon} \right] +$$
$$+ \frac{\eta_t D G \sqrt{N} \lambda^t}{\sqrt{1 - \beta_2}} + T_{2t} \tag{39}$$

$$\leq f(\mathbf{w}_t) - \eta_t \sum_{i=1}^{D} \frac{\nabla f(\mathbf{w}_t)_i^2}{\sqrt{\beta_2 \mathbf{v}_{t-1,i}} + \epsilon} + \eta_t \sum_{i=1}^{D} |\nabla f(\mathbf{w}_t)_i| \left| \mathbb{E}_{\boldsymbol{\pi}^*,\xi} \left[ \underbrace{\frac{\mathbf{g}_{t,i}}{\sqrt{\mathbf{v}_{t,i}} + \epsilon} - \frac{\mathbf{g}_{t,i}}{\sqrt{\beta_2 \mathbf{v}_{t-1,i}} + \epsilon}}_{T_{1t}} \right] \right| +$$
$$+ \frac{\eta_t D G \sqrt{N} \lambda^t}{\sqrt{1 - \beta_2}} + T_{2t}. \tag{40}$$

From [25] we have that the $T_{1t}$ term can be upper bounded by

$$T_{1t} \leq \frac{\sqrt{1 - \beta_2} \mathbf{g}_{t,i}^2}{(\sqrt{\beta_2 \mathbf{v}_{t-1,i}} + \epsilon)\epsilon}. \tag{41}$$

Therefore, taking this into account and with the additional assumption that each gradient coordinate is upper bounded by $G$, we have that

$$\leq f(\mathbf{w}_t) - \eta_t \sum_{i=1}^{D} \frac{\nabla f(\mathbf{w}_t)_i^2}{\sqrt{\beta_2 \mathbf{v}_{t-1,i}} + \epsilon} + \frac{G \sqrt{1 - \beta_2} \eta_t}{\epsilon} \sum_{i=1}^{D} \mathbb{E}_{\boldsymbol{\pi}^*,\xi} \left[ \frac{\mathbf{g}_{t,i}^2}{\sqrt{\beta_2 \mathbf{v}_{t-1,i}} + \epsilon} \right] +$$
$$+ \frac{\eta_t D G \sqrt{N} \lambda^t}{\sqrt{1 - \beta_2}} + T_{2t}. \tag{42}$$

Furthermore, due to the positivity of $\sqrt{\mathbf{v}_t}$ and $\epsilon$ we also have that

$$\sqrt{\mathbf{v}_{t,i}} + \epsilon \geq \epsilon \qquad \rightarrow \qquad (\sqrt{\mathbf{v}_{t,i}} + \epsilon)^2 \geq \epsilon(\sqrt{\mathbf{v}_{t,i}} + \epsilon). \tag{43}$$

Therefore, we have that the the $T_{2t}$ term can be upper bounded by

$$T_{2t} \leq \frac{L \eta_t^2}{2\epsilon} \sum_{i=1}^{D} \mathbb{E}_{\boldsymbol{\pi}^*,\xi} \left[ \frac{\mathbf{g}_{t,i}^2}{\sqrt{\mathbf{v}_{t,i}} + \epsilon} \right] + \frac{L \eta_t^2 D \sqrt{N} \lambda^t}{2(1 - \beta_2)} \tag{44}$$

$$\leq \frac{L \eta_t^2}{2\epsilon} \sum_{i=1}^{D} \mathbb{E}_{\boldsymbol{\pi}^*,\xi} \left[ \frac{\mathbf{g}_{t,i}^2}{\sqrt{\beta_2 \mathbf{v}_{t-1,i}} + \epsilon} \right] + \frac{L \eta_t^2 D \sqrt{N} \lambda^t}{2(1 - \beta_2)}, \tag{45}$$

since $\mathbf{v}_{t,i} = \beta_2 \mathbf{v}_{t-1,i} + (1 - \beta_2) \mathbf{g}_t^2 \geq \beta_2 \mathbf{v}_{t-1,i}$. Putting everything together, we have that

$$\leq f(\mathbf{w}_t) - \eta_t \sum_{i=1}^{D} \frac{\nabla f(\mathbf{w}_t)_i^2}{\sqrt{\beta_2 \mathbf{v}_{t-1,i}} + \epsilon} + \frac{G \sqrt{1 - \beta_2} \eta_t}{\epsilon} \sum_{i=1}^{D} \mathbb{E}_{\boldsymbol{\pi}^*,\xi} \left[ \frac{\mathbf{g}_{t,i}^2}{\sqrt{\beta_2 \mathbf{v}_{t-1,i}} + \epsilon} \right] +$$
$$+ \frac{L \eta_t^2}{2\epsilon} \sum_{i=1}^{D} \mathbb{E}_{\boldsymbol{\pi}^*,\xi} \left[ \frac{\mathbf{g}_{t,i}^2}{\sqrt{\beta_2 \mathbf{v}_{t-1,i}} + \epsilon} \right] + \left( \frac{L \eta_t^2}{2(1 - \beta_2)} + \frac{\eta_t G}{\sqrt{1 - \beta_2}} \right) D \sqrt{N} \lambda^t. \tag{46}$$

Now we have one more assumption, namely that the gradient variance of each dimension is bounded by $\sigma_i^2$. In this way, we have that $\mathbb{V}[\mathbf{g}_{t,i}] = \mathbb{E}[\mathbf{g}_{t,i}^2] - \mathbb{E}[\mathbf{g}_{t,i}]^2 \leq \sigma_i^2 \rightarrow \mathbb{E}[\mathbf{g}_{t,i}^2] \leq \sigma_i^2 + \mathbb{E}[\mathbf{g}_{t,i}]^2$.

Therefore, we have that

$$\leq f(\mathbf{w}_t) - \eta_t \sum_{i=1}^{D} \frac{\nabla f(\mathbf{w}_t)_i^2}{\sqrt{\beta_2 \mathbf{v}_{t-1,i}} + \epsilon} + \frac{G\sqrt{1-\beta_2}\eta_t}{\epsilon} \sum_{i=1}^{D} \frac{\nabla f(\mathbf{w}_t)_i^2 + \sigma_i^2}{\sqrt{\beta_2 \mathbf{v}_{t-1,i}} + \epsilon}$$

$$+ \frac{L\eta_t^2}{2\epsilon} \sum_{i=1}^{D} \frac{\nabla f(\mathbf{w}_t)_i^2 + \sigma_i^2}{\sqrt{\beta_2 \mathbf{v}_{t-1,i}} + \epsilon} + \left( \frac{L\eta_t^2}{2(1-\beta_2)} + \frac{\eta_t G}{\sqrt{1-\beta_2}} \right) D\sqrt{N}\lambda^t \qquad (47)$$

$$= f(\mathbf{w}_t) - \left( \eta_t - \frac{\eta_t G\sqrt{1-\beta_2}}{\epsilon} - \frac{L\eta_t^2}{2\epsilon} \right) \sum_{i=1}^{D} \frac{\nabla f(\mathbf{w}_t)_i^2}{\sqrt{\beta_2 \mathbf{v}_{t-1,i}} + \epsilon}$$

$$+ \left( \frac{\eta_t G\sqrt{1-\beta_2}}{\epsilon} + \frac{L\eta_t^2}{2\epsilon} \right) \sum_{i=1}^{D} \frac{\sigma_i^2}{\sqrt{\beta_2 \mathbf{v}_{t-1,i}} + \epsilon} + \left( \frac{L\eta_t^2}{2(1-\beta_2)} + \frac{\eta_t G}{\sqrt{1-\beta_2}} \right) D\sqrt{N}\lambda^t. \qquad (48)$$

There are some further assumptions taken from [25], namely that $\eta_t, \beta_2, \epsilon$ are chosen such that

$$\frac{L\eta_t}{2\epsilon} \leq \frac{1}{4}, \qquad \frac{G\sqrt{1-\beta_2}}{\epsilon} \leq \frac{1}{4}. \qquad (49)$$

Using these, we have that

$$\leq f(\mathbf{w}_t) - \left( \eta_t - \frac{\eta_t}{4} - \frac{\eta_t}{4} \right) \sum_{i=1}^{D} \frac{\nabla f(\mathbf{w}_t)_i^2}{\sqrt{\beta_2 \mathbf{v}_{t-1,i}} + \epsilon}$$

$$+ \left( \frac{\eta_t G\sqrt{1-\beta_2}}{\epsilon} + \frac{L\eta_t^2}{2\epsilon} \right) \sum_{i=1}^{D} \frac{\sigma_i^2}{\sqrt{\beta_2 \mathbf{v}_{t-1,i}} + \epsilon} + \left( \frac{L\eta_t^2}{2(1-\beta_2)} + \frac{\eta_t G}{\sqrt{1-\beta_2}} \right) D\sqrt{N}\lambda^t \qquad (50)$$

$$= f(\mathbf{w}_t) - \frac{\eta_t}{2} \sum_{i=1}^{D} \frac{\nabla f(\mathbf{w}_t)_i^2}{\sqrt{\beta_2 \mathbf{v}_{t-1,i}} + \epsilon}$$

$$+ \left( \frac{\eta_t G\sqrt{1-\beta_2}}{\epsilon} + \frac{L\eta_t^2}{2\epsilon} \right) \sum_{i=1}^{D} \frac{\sigma_i^2}{\sqrt{\beta_2 \mathbf{v}_{t-1,i}} + \epsilon} + \left( \frac{L\eta_t^2}{2(1-\beta_2)} + \frac{\eta_t G}{\sqrt{1-\beta_2}} \right) D\sqrt{N}\lambda^t, \qquad (51)$$

and due to $0 \leq \mathbf{v}_{t,i} \leq G^2$ and defining $\sigma^2 = \sum_{i=1}^{D} \sigma_i^2$ we have that

$$\leq f(\mathbf{w}_t) - \frac{\eta_t}{2(\sqrt{\beta_2}G + \epsilon)} \|\nabla f(\mathbf{w}_t)\|^2 + \left( \frac{\eta_t G\sqrt{1-\beta_2}}{\epsilon^2} + \frac{L\eta_t^2}{2\epsilon^2} \right) \sigma^2$$

$$+ \left( \frac{L\eta_t^2}{2(1-\beta_2)} + \frac{\eta_t G}{\sqrt{1-\beta_2}} \right) D\sqrt{N}\lambda^t. \qquad (52)$$

Therefore,

$$\mathbb{E}_{\boldsymbol{\pi}_t}[f(\mathbf{w}_{t+1})] - f(\mathbf{w}_t) \leq -\frac{\eta_t}{2(\sqrt{\beta_2}G + \epsilon)} \|\nabla f(\mathbf{w}_t)\|^2 + \left( \frac{\eta_t G\sqrt{1-\beta_2}}{\epsilon^2} + \frac{L\eta_t^2}{2\epsilon^2} \right) \sigma^2 +$$

$$+ \left( \frac{L\eta_t^2}{2(1-\beta_2)} + \frac{\eta_t G}{\sqrt{1-\beta_2}} \right) D\sqrt{N}\lambda^t. \qquad (53)$$

Now by setting $\eta_t = \eta$ and taking a telescoping sum, *i.e.*, taking the expectation over $\boldsymbol{\pi}_t$ for all $t$ we have that

$$\sum_t \mathbb{E}_{\boldsymbol{\pi}_t,\xi}[f(\mathbf{w}_{t+1})] - \mathbb{E}_{\boldsymbol{\pi}_{t-1},\xi}[f(\mathbf{w}_t)] \leq -\frac{\eta}{2(\sqrt{\beta_2}G + \epsilon)} \sum_{t=1}^{T} \|\nabla f(\mathbf{w}_t)\|^2$$

$$+ \left( \frac{\eta G\sqrt{1-\beta_2}}{\epsilon^2} + \frac{L\eta^2}{2\epsilon^2} \right) T\sigma^2 + \left( \frac{L\eta^2}{2(1-\beta_2)} + \frac{\eta G}{\sqrt{1-\beta_2}} \right) D\sqrt{N} \sum_{t=1}^{T} \lambda^t \qquad (54)$$

$$\mathbb{E}_{\boldsymbol{\pi}_T,\xi}[f(\mathbf{w}_{T+1})] - f(\mathbf{w}_1) \leq -\frac{\eta}{2(\sqrt{\beta_2}G + \epsilon)} \sum_{t=1}^{T} \|\nabla f(\mathbf{w}_t)\|^2 + \left( \frac{\eta G\sqrt{1-\beta_2}}{\epsilon^2} + \frac{L\eta^2}{2\epsilon^2} \right) T\sigma^2$$

$$+ \left( \frac{L\eta^2}{2(1-\beta_2)} + \frac{\eta G}{\sqrt{1-\beta_2}} \right) D\sqrt{N} \sum_{t=1}^{T} \lambda^t. \qquad (55)$$

Rearranging the inequality and assuming that $f(\mathbf{w}^*) \leq f(\mathbf{w}_t) \forall t$ (*i.e.*, $\mathbf{w}^*$ is the optimum) we have that

$$\frac{\eta}{2(\sqrt{\beta_2}G + \epsilon)} \sum_{t=1}^{T} \|\nabla f(\mathbf{w}_t)\|^2 \leq f(\mathbf{w}_1) - \mathop{\mathbb{E}}_{\boldsymbol{\pi}_T, \xi}[f(\mathbf{w}_{T+1})] + \left(\frac{\eta G\sqrt{1-\beta_2}}{\epsilon^2} + \frac{L\eta^2}{2\epsilon^2}\right) T\sigma^2 +$$

$$+ \left(\frac{L\eta^2}{2(1-\beta_2)} + \frac{\eta G}{\sqrt{1-\beta_2}}\right) D\sqrt{N} \sum_{t=1}^{T} \lambda^t \tag{56}$$

$$\frac{1}{T} \sum_{t=1}^{T} \|\nabla f(\mathbf{w}_t)\|^2 \leq 2(\sqrt{\beta_2}G + \epsilon) \left(\frac{f(\mathbf{w}_1) - \mathbb{E}_{\boldsymbol{\pi}_T, \xi}[f(\mathbf{w}_{T+1})]}{\eta T} + \left(\frac{G\sqrt{1-\beta_2}}{\epsilon^2} + \frac{L\eta}{2\epsilon^2}\right)\sigma^2 \right.$$

$$\left. + \left(\frac{L\eta}{2(1-\beta_2)} + \frac{G}{\sqrt{1-\beta_2}}\right) D\sqrt{N} \frac{1}{T} \sum_{t=1}^{T} \lambda^t \right) \tag{57}$$

$$\frac{1}{T} \sum_{t=1}^{T} \|\nabla f(\mathbf{w}_t)\|^2 \leq 2(\sqrt{\beta_2}G + \epsilon) \left(\frac{f(\mathbf{w}_1) - f(\mathbf{w}^*)}{\eta T} + \left(\frac{G\sqrt{1-\beta_2}}{\epsilon^2} + \frac{L\eta}{2\epsilon^2}\right)\sigma^2 \right.$$

$$\left. + \left(\frac{L\eta}{2(1-\beta_2)} + \frac{G}{\sqrt{1-\beta_2}}\right) D\sqrt{N} \frac{1}{T} \sum_{t=1}^{T} \lambda^t \right). \tag{58}$$

Furthermore, since $\lambda < 1$, we have that the $\sum_t \lambda^t$ forms a geometric series, and therefore it is upper bounded by

$$\sum_{t=0}^{T} \lambda^t \leq \frac{1}{1-\lambda} \quad \rightarrow \quad 1 + \sum_{t=1}^{T} \lambda^t \leq \frac{1}{1-\lambda} \quad \rightarrow \quad \sum_{t=1}^{T} \lambda^t \leq \frac{\lambda}{1-\lambda}. \tag{59}$$

Therefore, the above can be simplified to

$$\frac{1}{T} \sum_{t=1}^{T} \|\nabla f(\mathbf{w}_t)\|^2 \leq 2(\sqrt{\beta_2}G + \epsilon) \left(\frac{f(\mathbf{w}_1) - f(\mathbf{w}^*)}{\eta T} + \left(\frac{G\sqrt{1-\beta_2}}{\epsilon^2} + \frac{L\eta}{2\epsilon^2}\right)\sigma^2 \right.$$

$$\left. + \frac{\lambda}{T(1-\lambda)} \left(\frac{L\eta}{2(1-\beta_2)} + \frac{G}{\sqrt{1-\beta_2}}\right) D\sqrt{N} \right). \tag{60}$$

Therefore, we have that

$$\mathbb{E}[\|\nabla f(\mathbf{w}_a)\|^2] \leq O\left(\frac{f(\mathbf{w}_1) - f(\mathbf{w}^*)}{\eta T} + \sigma^2 + \frac{\lambda D\sqrt{N}}{(1-\lambda)T}\right), \tag{61}$$

where $\mathbf{w}_a$ is randomly chosen iterate from $\mathbf{w}_1, \ldots, \mathbf{w}_T$. This ends up being a result similar to the one at [25], except with an additional term that depends sublinearly on the number of nodes ($N$) and decays with the number of iterations $T$.

### A.3 Adding compression

To use adaptive optimizers effectively in our setting, it show in the main text that it is useful to quantize the second moment. We will have to take this into account in the proof. We begin the proof in a similar manner; the updates when the second moment is quantized are given as

$$\mathbf{w}_{t+1,i} = \mathbf{w}_{t,i} - \eta_t \frac{\mathbf{g}_{t,i}}{\sqrt{\mathbf{v}_{t,i}} + \epsilon}, \qquad \mathbf{w}_{t+1,i} - \mathbf{w}_{t,i} = -\eta_t \frac{\mathbf{g}_{t,i}}{\sqrt{\mathbf{v}_{t,i}} + \epsilon}, \tag{62}$$

where

$$\mathbf{v}_{t,i} = \beta_2(\mathbf{v}_{t-1,i} + \mathbf{r}_{t-1,i}) + (1 - \beta_2)\mathbf{g}_{t,i}^2. \tag{63}$$

$\mathbf{r}_{t-1,i}$ is the quantization noise on the second moment, *i.e.*, $\mathbf{r}_{t-1,i} = Q(\mathbf{v}_{t-1,i}) - \mathbf{v}_{t-1,i}$, with $Q(\mathbf{v})$ representing the quantized second moment. We then assume a L-Lipschitz loss function and proceed

by:

$$f(\mathbf{w}_{t+1}) \le f(\mathbf{w}_t) + \nabla f(\mathbf{w}_t)^T(\mathbf{w}_{t+1} - \mathbf{w}_t) + \frac{L}{2}\|\mathbf{w}_{t+1} - \mathbf{w}_t\|^2 \tag{64}$$

$$= f(\mathbf{w}_t) - \eta_t \sum_{i=1}^{D} \nabla f(\mathbf{w}_t)_i \frac{\mathbf{g}_{t,i}}{\sqrt{\mathbf{v}_{t,i}} + \epsilon} + \frac{L\eta_t^2}{2}\sum_{i=1}^{D}\frac{\mathbf{g}_{t,i}^2}{\left(\sqrt{\mathbf{v}_{t,i}} + \epsilon\right)^2}. \tag{65}$$

We then take an expectation over the distribution of the chain at timestep $t$, $\boldsymbol{\pi}_t$, and the data-samples $\xi$ given that we are at $\mathbf{w}_t$ along with an expectation over the quantization noise of round $t-1$, $\mathbf{r}_{t-1}$.

$$\mathop{\mathbb{E}}_{\boldsymbol{\pi}_t, \mathbf{r}_{t-1}, \xi}[f(\mathbf{w}_{t+1})] \le f(\mathbf{w}_t) - \eta_t \sum_{i=1}^{D} \nabla f(\mathbf{w}_t)_i \mathop{\mathbb{E}}_{\boldsymbol{\pi}_t, \mathbf{r}_{t-1}, \xi}\left[\frac{\mathbf{g}_{t,i}}{\sqrt{\mathbf{v}_{t,i}} + \epsilon}\right]$$

$$+ \frac{L\eta_t^2}{2}\sum_{i=1}^{D} \mathop{\mathbb{E}}_{\boldsymbol{\pi}_t, \mathbf{r}_{t-1}, \xi}\left[\frac{\mathbf{g}_{t,i}^2}{\left(\sqrt{\mathbf{v}_{t,i}} + \epsilon\right)^2}\right], \tag{66}$$

and we then apply Lemma 4.1 to express the expectations in the r.h.s. to ones over the stationary distribution at timestep $t$, $\boldsymbol{\pi}^*$.

$$\mathop{\mathbb{E}}_{\boldsymbol{\pi}_t, \mathbf{r}_{t-1}, \xi}[f(\mathbf{w}_{t+1})] \le f(\mathbf{w}_t) - \eta_t \sum_{i=1}^{D} \nabla f(\mathbf{w}_t)_i \mathop{\mathbb{E}}_{\boldsymbol{\pi}^*, \mathbf{r}_{t-1}, \xi}\left[\frac{\mathbf{g}_{t,i}}{\sqrt{\mathbf{v}_{t,i}} + \epsilon}\right] + \frac{\eta_t D G \sqrt{N}\lambda^t}{\sqrt{1 - \beta_2}}$$

$$+ \underbrace{\frac{L\eta_t^2}{2}\sum_{i=1}^{D} \mathop{\mathbb{E}}_{\boldsymbol{\pi}^*, \mathbf{r}_{t-1}, \xi}\left[\frac{\mathbf{g}_{t,i}^2}{\left(\sqrt{\mathbf{v}_{t,i}} + \epsilon\right)^2}\right] + \frac{L\eta_t^2 D\sqrt{N}\lambda^t}{2(1 - \beta_2)}}_{T_{2t}}, \tag{67}$$

and then proceed in a similar manner as before.

$$\le f(\mathbf{w}_t) - \eta_t \sum_{i=1}^{D} \nabla f(\mathbf{w}_t)_i \mathop{\mathbb{E}}_{\boldsymbol{\pi}^*, \mathbf{r}_{t-1}, \xi}\left[\frac{\mathbf{g}_{t,i}}{\sqrt{\mathbf{v}_{t,i}} + \epsilon} + \frac{\mathbf{g}_{t,i}}{\sqrt{\beta_2 \mathbf{v}_{t-1,i}} + \epsilon} - \frac{\mathbf{g}_{t,i}}{\sqrt{\beta_2 \mathbf{v}_{t-1,i}} + \epsilon}\right]$$

$$+ \frac{\eta_t D G \sqrt{N}\lambda^t}{\sqrt{1 - \beta_2}} + T_{2t}, \tag{68}$$

and since $\mathbb{E}_{\boldsymbol{\pi}^*, \mathbf{r}_{t-1}, \xi}[\mathbf{g}_t] = \mathbb{E}_{\boldsymbol{\pi}^*, \xi}[\mathbf{g}_t] = \nabla f(\mathbf{w}_t)$ (as the quantization noise on timestep $t-1$ does not affect the gradient at timestep $t$ once we condition on $\mathbf{w}_t$),

$$= f(\mathbf{w}_t) - \eta_t \sum_{i=1}^{D} \nabla f(\mathbf{w}_t)_i \left(\frac{\nabla f(\mathbf{w}_t)_i}{\sqrt{\beta_2 \mathbf{v}_{t-1,i}} + \epsilon} + \mathop{\mathbb{E}}_{\boldsymbol{\pi}^*, \mathbf{r}_{t-1}, \xi}\left[\frac{\mathbf{g}_{t,i}}{\sqrt{\mathbf{v}_{t,i}} + \epsilon} - \frac{\mathbf{g}_{t,i}}{\sqrt{\beta_2 \mathbf{v}_{t-1,i}} + \epsilon}\right]\right)$$

$$+ \frac{\eta_t D G \sqrt{N}\lambda^t}{\sqrt{1 - \beta_2}} + T_{2t} \tag{69}$$

$$= f(\mathbf{w}_t) - \eta_t \sum_{i=1}^{D} \frac{\nabla f(\mathbf{w}_t)_i^2}{\sqrt{\beta_2 \mathbf{v}_{t-1,i}} + \epsilon} - \eta_t \sum_{i=1}^{D} \nabla f(\mathbf{w}_t)_i \mathop{\mathbb{E}}_{\boldsymbol{\pi}^*, \mathbf{r}_{t-1}, \xi}\left[\frac{\mathbf{g}_{t,i}}{\sqrt{\mathbf{v}_{t,i}} + \epsilon} - \frac{\mathbf{g}_{t,i}}{\sqrt{\beta_2 \mathbf{v}_{t-1,i}} + \epsilon}\right]$$

$$+ \frac{\eta_t D G \sqrt{N}\lambda^t}{\sqrt{1 - \beta_2}} + T_{2t} \tag{70}$$

$$\le f(\mathbf{w}_t) - \eta_t \sum_{i=1}^{D} \frac{\nabla f(\mathbf{w}_t)_i^2}{\sqrt{\beta_2 \mathbf{v}_{t-1,i}} + \epsilon}$$

$$+ \eta_t \sum_{i=1}^{D} |\nabla f(\mathbf{w}_t)_i| \left|\mathop{\mathbb{E}}_{\boldsymbol{\pi}^*, \mathbf{r}_{t-1}, \xi}\left[\underbrace{\frac{\mathbf{g}_{t,i}}{\sqrt{\mathbf{v}_{t,i}} + \epsilon} - \frac{\mathbf{g}_{t,i}}{\sqrt{\beta_2 \mathbf{v}_{t-1,i}} + \epsilon}}_{T_{1t}}\right]\right| + \frac{\eta_t D G \sqrt{N}\lambda^t}{\sqrt{1 - \beta_2}} + T_{2t}. \tag{71}$$

As we have quantization of the second moments, the bound from [25] doesn't apply. However, we can construct the following upper bound:

$$T_{1t} \leq |\mathbf{g}_{t,i}| \left| \frac{1}{\sqrt{\mathbf{v}_{t,i}} + \epsilon} - \frac{1}{\sqrt{\beta_2 \mathbf{v}_{t-1,i}} + \epsilon} \right| \tag{72}$$

$$= |\mathbf{g}_{t,i}| \left| \frac{\sqrt{\mathbf{v}_{t,i}} - \sqrt{\beta_2 \mathbf{v}_{t-1,i}}}{(\sqrt{\mathbf{v}_t} + \epsilon)(\sqrt{\beta_2 \mathbf{v}_{t-1,i}} + \epsilon)} \right| \tag{73}$$

$$= \frac{|\mathbf{g}_{t,i}|}{(\sqrt{\mathbf{v}_t} + \epsilon)(\sqrt{\beta_2 \mathbf{v}_{t-1,i}} + \epsilon)} \left| \frac{\mathbf{v}_{t,i} - \beta_2 \mathbf{v}_{t-1,i}}{\sqrt{\mathbf{v}_{t,i}} + \sqrt{\beta_2 \mathbf{v}_{t-1,i}}} \right| \tag{74}$$

$$= \frac{|\mathbf{g}_{t,i}|}{(\sqrt{\mathbf{v}_{t,i}} + \epsilon)(\sqrt{\beta_2 \mathbf{v}_{t-1,i}} + \epsilon)(\sqrt{\mathbf{v}_{t,i}} + \sqrt{\beta_2 \mathbf{v}_{t-1,i}})} |(1 - \beta_2)\mathbf{g}_{t,i}^2 + \beta_2 \mathbf{r}_{t-1,i}| \tag{75}$$

$$\leq \frac{(1 - \beta_2)|\mathbf{g}_{t,i}|\mathbf{g}_{t,i}^2}{(\sqrt{\mathbf{v}_{t,i}} + \epsilon)(\sqrt{\beta_2 \mathbf{v}_{t-1,i}} + \epsilon)(\sqrt{\mathbf{v}_{t,i}} + \sqrt{\beta_2 \mathbf{v}_{t-1,i}})}$$

$$+ \frac{\beta_2 |\mathbf{g}_{t,i}||\mathbf{r}_{t-1,i}|}{(\sqrt{\mathbf{v}_{t,i}} + \epsilon)(\sqrt{\beta_2 \mathbf{v}_{t-1,i}} + \epsilon)(\sqrt{\mathbf{v}_{t,i}} + \sqrt{\beta_2 \mathbf{v}_{t-1,i}})}. \tag{76}$$

With our assumptions, we can show that

$$\frac{|\mathbf{g}_{t,i}|}{\sqrt{\mathbf{v}_{t,i}} + \epsilon} = \frac{|\mathbf{g}_{t,i}|}{\sqrt{\beta_2(\mathbf{v}_{t-1,i} + \mathbf{r}_{t-1,i}) + (1 - \beta_2)\mathbf{g}_{t,i}^2} + \epsilon} \leq \frac{|\mathbf{g}_{t,i}|}{\sqrt{(1 - \beta_2)\mathbf{g}_{t,i}^2}} = \frac{1}{\sqrt{1 - \beta_2}} \tag{77}$$

$$\frac{|\mathbf{g}_{t,i}|}{\sqrt{\mathbf{v}_{t,i}} + \sqrt{\beta_2 \mathbf{v}_{t-1,i}}} = \frac{|\mathbf{g}_{t,i}|}{\sqrt{\beta_2(\mathbf{v}_{t-1,i} + \mathbf{r}_{t-1,i}) + (1 - \beta_2)\mathbf{g}_{t,i}^2} + \sqrt{\beta_2 \mathbf{v}_{t-1}}}$$

$$\leq \frac{|\mathbf{g}_{t,i}|}{\sqrt{(1 - \beta_2)\mathbf{g}_{t,i}^2}} = \frac{1}{\sqrt{1 - \beta_2}}, \tag{78}$$

since the minimum value of $\mathbf{v}_{t-1,i} + \mathbf{r}_{t-1,i} \geq 0$, *i.e.*, the quantized second moment cannot be negative. Thus, using these upper bounds we have that

$$T_{1t} \leq \frac{\sqrt{1 - \beta_2}\mathbf{g}_{t,i}^2}{(\sqrt{\mathbf{v}_{t,i}} + \epsilon)(\sqrt{\beta_2 \mathbf{v}_{t-1,i}} + \epsilon)} + \frac{\beta_2 |\mathbf{r}_{t-1,i}|}{\sqrt{1 - \beta_2}(\sqrt{\beta_2 \mathbf{v}_{t-1,i}} + \epsilon)(\sqrt{\mathbf{v}_{t,i}} + \sqrt{\beta_2 \mathbf{v}_{t-1,i}})} \tag{79}$$

$$\leq \frac{\sqrt{1 - \beta_2}\mathbf{g}_{t,i}^2}{\epsilon(\sqrt{\beta_2 \mathbf{v}_{t-1,i}} + \epsilon)} + \frac{\beta_2 |\mathbf{r}_{t-1,i}|}{\sqrt{1 - \beta_2}(\sqrt{\beta_2 \mathbf{v}_{t-1,i}} + \epsilon)(\sqrt{\mathbf{v}_{t,i}} + \sqrt{\beta_2 \mathbf{v}_{t-1,i}})}, \tag{80}$$

and due to our assumptions on the quantization noise, i.e., Assumption 4 we have that

$$\frac{|\mathbf{r}_{t-1,i}|}{\sqrt{\beta_2 \mathbf{v}_{t-1,i}} + \epsilon} \leq \frac{|\mathbf{r}_{t-1,i}|}{\sqrt{\beta_2 \mathbf{v}_{t-1,i}}} \leq \frac{q_{t-1}\mathbf{v}_{t-1,i}}{\sqrt{\beta_2}\sqrt{\mathbf{v}_{t-1,i}}} = \frac{q_{t-1}\sqrt{\mathbf{v}_{t-1,i}}}{\sqrt{\beta_2}}. \tag{81}$$

We thus end up with

$$T_{1t} \leq \frac{\sqrt{1 - \beta_2}\mathbf{g}_{t,i}^2}{\epsilon(\sqrt{\beta_2 \mathbf{v}_{t-1,i}} + \epsilon)} + \frac{q_{t-1}\sqrt{\beta_2 \mathbf{v}_{t-1,i}}}{\sqrt{1 - \beta_2}(\sqrt{\mathbf{v}_{t,i}} + \sqrt{\beta_2 \mathbf{v}_{t-1,i}})} \tag{82}$$

$$\leq \frac{\sqrt{1 - \beta_2}\mathbf{g}_{t,i}^2}{\epsilon(\sqrt{\beta_2 \mathbf{v}_{t-1,i}} + \epsilon)} + \frac{q_{t-1}}{\sqrt{1 - \beta_2}}. \tag{83}$$

Therefore, taking the bound on $T_{1t}$ into account and with the additional assumption that each gradient coordinate is upper bounded by $G$, we have that

$$\leq f(\mathbf{w}_t) - \eta_t \sum_{i=1}^{D} \frac{\nabla f(\mathbf{w}_t)_i^2}{\sqrt{\beta_2 \mathbf{v}_{t-1,i}} + \epsilon} + \frac{G\sqrt{1 - \beta_2}\eta_t}{\epsilon} \sum_{i=1}^{D} \mathop{\mathbb{E}}_{\pi^*,\xi} \left[ \frac{\mathbf{g}_{t,i}^2}{\sqrt{\beta_2 \mathbf{v}_{t-1,i}} + \epsilon} \right]$$

$$+ \frac{DG\eta_t q_{t-1}}{\sqrt{1 - \beta_2}} + \frac{\eta_t DG\sqrt{N}\lambda^t}{\sqrt{1 - \beta_2}} + T_{2t} \tag{84}$$

Let us now take a look at the second term $T_{2t}$. For that term we have that

$$T_{2t} = \frac{L\eta_t^2}{2} \sum_{i=1}^{D} \mathop{\mathbb{E}}_{\boldsymbol{\pi}^*, \mathbf{r}_{t-1}, \xi} \left[ \frac{\mathbf{g}_{t,i}^2}{\left(\sqrt{\mathbf{v}_{t,i}} + \epsilon\right)^2} \right] + \frac{L\eta_t^2 D\sqrt{N}\lambda^t}{2(1-\beta_2)} \tag{85}$$

$$\leq \frac{L\eta_t^2}{2\epsilon} \sum_{i=1}^{D} \mathop{\mathbb{E}}_{\boldsymbol{\pi}^*, \mathbf{r}_{t-1}} \left[ \frac{\mathbf{g}_{t,i}^2}{\sqrt{\mathbf{v}_{t,i}} + \epsilon} \right] + \frac{L\eta_t^2 D\sqrt{N}\lambda^t}{2(1-\beta_2)} \tag{86}$$

$$= \frac{L\eta_t^2}{2\epsilon} \sum_{i=1}^{D} \mathop{\mathbb{E}}_{\boldsymbol{\pi}^*, \mathbf{r}_{t-1}, \xi} \left[ \frac{\mathbf{g}_{t,i}^2}{\sqrt{\mathbf{v}_{t,i}} + \epsilon} + \frac{\mathbf{g}_{t,i}^2}{\sqrt{\beta_2 \mathbf{v}_{t-1,i}} + \epsilon} - \frac{\mathbf{g}_{t,i}^2}{\sqrt{\beta_2 \mathbf{v}_{t-1,i}} + \epsilon} \right] + \frac{L\eta_t^2 D\sqrt{N}\lambda^t}{2(1-\beta_2)} \tag{87}$$

$$= \frac{L\eta_t^2}{2\epsilon} \sum_{i=1}^{D} \mathop{\mathbb{E}}_{\boldsymbol{\pi}^*, \xi} \left[ \frac{\mathbf{g}_{t,i}^2}{\sqrt{\beta_2 \mathbf{v}_{t-1,i}} + \epsilon} \right] + \frac{L\eta_t^2 D\sqrt{N}\lambda^t}{2(1-\beta_2)} +$$

$$+ \frac{L\eta_t^2}{2\epsilon} \sum_{i=1}^{D} \mathop{\mathbb{E}}_{\boldsymbol{\pi}^*, \mathbf{r}_{t-1}, \xi} \left[ \underbrace{\frac{\mathbf{g}_{t,i}^2}{\sqrt{\mathbf{v}_{t,i}} + \epsilon} - \frac{\mathbf{g}_{t,i}^2}{\sqrt{\beta_2 \mathbf{v}_{t-1,i}} + \epsilon}}_{T_{3t}} \right]. \tag{88}$$

We now turn to bounding $T_{3t}$. We have that

$$T_{3t} \leq \mathbf{g}_{t,i}^2 \left| \frac{1}{\sqrt{\mathbf{v}_{t,i}} + \epsilon} - \frac{1}{\sqrt{\beta_2 \mathbf{v}_{t-1,i}} + \epsilon} \right| \tag{89}$$

$$= \mathbf{g}_{t,i}^2 \left| \frac{\sqrt{\mathbf{v}_{t,i}} - \sqrt{\beta_2 \mathbf{v}_{t-1,i}}}{(\sqrt{\mathbf{v}_{t,i}} + \epsilon)(\sqrt{\beta_2 \mathbf{v}_{t-1,i}} + \epsilon)} \right| \tag{90}$$

$$= \mathbf{g}_{t,i}^2 \left| \frac{(\sqrt{\mathbf{v}_{t,i}} - \sqrt{\beta_2 \mathbf{v}_{t-1,i}})(\sqrt{\mathbf{v}_{t,i}} + \sqrt{\beta_2 \mathbf{v}_{t-1,i}})}{(\sqrt{\mathbf{v}_{t,i}} + \epsilon)(\sqrt{\beta_2 \mathbf{v}_{t-1,i}} + \epsilon)(\sqrt{\mathbf{v}_{t,i}} + \sqrt{\beta_2 \mathbf{v}_{t-1,i}})} \right| \tag{91}$$

$$= \mathbf{g}_{t,i}^2 \left| \frac{\mathbf{v}_{t,i} - \beta_2 \mathbf{v}_{t-1,i}}{(\sqrt{\mathbf{v}_{t,i}} + \epsilon)(\sqrt{\beta_2 \mathbf{v}_{t-1,i}} + \epsilon)(\sqrt{\mathbf{v}_{t,i}} + \sqrt{\beta_2 \mathbf{v}_{t-1,i}})} \right| \tag{92}$$

$$= \frac{\mathbf{g}_{t,i}^2}{(\sqrt{\mathbf{v}_{t,i}} + \epsilon)(\sqrt{\beta_2 \mathbf{v}_{t-1,i}} + \epsilon)} \left| \frac{(1-\beta_2)\mathbf{g}_{t,i}^2 + \beta_2 \mathbf{r}_{t-1,i}}{\sqrt{\mathbf{v}_{t,i}} + \sqrt{\beta_2 \mathbf{v}_{t-1,i}}} \right| \tag{93}$$

$$\leq \frac{(1-\beta_2)\mathbf{g}_{t,i}^4}{(\sqrt{\mathbf{v}_{t,i}} + \epsilon)(\sqrt{\beta_2 \mathbf{v}_{t-1,i}} + \epsilon)(\sqrt{\mathbf{v}_{t,i}} + \sqrt{\beta_2 \mathbf{v}_{t-1,i}})} +$$

$$+ \frac{\beta_2 \mathbf{g}_{t,i}^2 |\mathbf{r}_{t-1,i}|}{(\sqrt{\mathbf{v}_{t,i}} + \epsilon)(\sqrt{\beta_2 \mathbf{v}_{t-1,i}} + \epsilon)(\sqrt{\mathbf{v}_{t,i}} + \sqrt{\beta_2 \mathbf{v}_{t-1,i}})}. \tag{94}$$

For the first term, we have that

$$\frac{(1-\beta_2)\mathbf{g}_{t,i}^4}{(\sqrt{\mathbf{v}_{t,i}} + \epsilon)(\sqrt{\beta_2 \mathbf{v}_{t-1,i}} + \epsilon)(\sqrt{\mathbf{v}_{t,i}} + \sqrt{\beta_2 \mathbf{v}_{t-1,i}})} \leq \frac{\sqrt{1-\beta_2}|\mathbf{g}_{t,i}|\mathbf{g}_{t,i}^2}{(\sqrt{\mathbf{v}_{t,i}} + \epsilon)(\sqrt{\beta_2 \mathbf{v}_{t-1,i}} + \epsilon)} \tag{95}$$

$$\leq \frac{\mathbf{g}_{t,i}^2}{\sqrt{\beta_2 \mathbf{v}_{t-1,i}} + \epsilon}. \tag{96}$$

For the second term, we have that

$$\frac{\beta_2 \mathbf{g}_{t,i}^2 |\mathbf{r}_{t-1,i}|}{(\sqrt{\mathbf{v}_{t,i}} + \epsilon)(\sqrt{\beta_2 \mathbf{v}_{t-1,i}} + \epsilon)(\sqrt{\mathbf{v}_{t,i}} + \sqrt{\beta_2 \mathbf{v}_{t-1,i}})} \leq \frac{q_{t-1}\mathbf{g}_{t,i}^2}{\sqrt{\mathbf{v}_{t,i}} + \epsilon} \leq \frac{q_{t-1}|\mathbf{g}_{t,i}|}{\sqrt{1-\beta_2}} \leq \frac{q_{t-1}G}{\sqrt{1-\beta_2}}. \tag{97}$$

Thus, by putting everything together, we have that

$$T_{3t} \leq \frac{\mathbf{g}_{t,i}^2}{\sqrt{\beta_2 \mathbf{v}_{t-1,i}} + \epsilon} + \frac{q_{t-1}G}{\sqrt{1-\beta_2}}. \tag{98}$$

Therefore, we have that $T_{2t}$ can be bounded as follows

$$T_{2t} \leq \frac{L\eta_t^2}{2\epsilon} \sum_{i=1}^{D} \mathbb{E}_{\pi^*,\xi} \left[ \frac{\mathbf{g}_{t,i}^2}{\sqrt{\beta_2 \mathbf{v}_{t-1,i}} + \epsilon} \right] + \frac{L\eta_t^2 D\sqrt{N}\lambda^t}{2(1-\beta_2)}$$

$$+ \frac{L\eta_t^2}{2\epsilon} \sum_{i=1}^{D} \mathbb{E}_{\pi^*,\xi} \left[ \frac{\mathbf{g}_{t,i}^2}{\sqrt{\beta_2 \mathbf{v}_{t-1,i}} + \epsilon} + \frac{q_{t-1}G}{\sqrt{1-\beta_2}} \right] \tag{99}$$

$$= \frac{2L\eta_t^2}{2\epsilon} \sum_{i=1}^{D} \mathbb{E}_{\pi^*,\xi} \left[ \frac{\mathbf{g}_{t,i}^2}{\sqrt{\beta_2 \mathbf{v}_{t-1,i}} + \epsilon} \right] + \frac{L\eta_t^2 D\sqrt{N}\lambda^t}{2(1-\beta_2)} + \frac{LDGq_{t-1}\eta_t^2}{2\epsilon\sqrt{1-\beta_2}}. \tag{100}$$

Thus, putting everything together, we have that

$$\leq f(\mathbf{w}_t) - \eta_t \sum_{i=1}^{D} \frac{\nabla f(\mathbf{w}_t)_i^2}{\sqrt{\beta_2 \mathbf{v}_{t-1,i}} + \epsilon} + \frac{G\sqrt{1-\beta_2}\eta_t}{\epsilon} \sum_{i=1}^{D} \mathbb{E}_{\pi^*,\xi} \left[ \frac{\mathbf{g}_{t,i}^2}{\sqrt{\beta_2 \mathbf{v}_{t-1,i}} + \epsilon} \right]$$

$$+ \frac{DG\eta_t q_{t-1}}{\sqrt{1-\beta_2}} + \frac{2L\eta_t^2}{2\epsilon} \sum_{i=1}^{D} \mathbb{E}_{\pi^*,\xi} \left[ \frac{\mathbf{g}_{t,i}^2}{\sqrt{\beta_2 \mathbf{v}_{t-1,i}} + \epsilon} \right] + \frac{LDGq_{t-1}\eta_t^2}{2\epsilon\sqrt{1-\beta_2}} +$$

$$+ \frac{L\eta_t^2 D\sqrt{N}\lambda^t}{2(1-\beta_2)} + \frac{\eta_t DG\sqrt{N}\lambda^t}{\sqrt{1-\beta_2}} \tag{101}$$

$$\leq f(\mathbf{w}_t) - \eta_t \sum_{i=1}^{D} \frac{\nabla f(\mathbf{w}_t)_i^2}{\sqrt{\beta_2 \mathbf{v}_{t-1,i}} + \epsilon} + \frac{G\sqrt{1-\beta_2}\eta_t}{\epsilon} \sum_{i=1}^{D} \frac{\nabla f(\mathbf{w}_t)_i^2 + \sigma_i^2}{\sqrt{\beta_2 \mathbf{v}_{t-1,i}} + \epsilon}$$

$$+ \frac{DG\eta_t q_{t-1}}{\sqrt{1-\beta_2}} + \frac{2L\eta_t^2}{2\epsilon} \sum_{i=1}^{D} \frac{\nabla f(\mathbf{w}_t)_i^2 + \sigma_i^2}{\sqrt{\beta_2 \mathbf{v}_{t-1,i}} + \epsilon} + \frac{LDGq_{t-1}\eta_t^2}{2\epsilon\sqrt{1-\beta_2}} +$$

$$+ \frac{L\eta_t^2 D\sqrt{N}\lambda^t}{2(1-\beta_2)} + \frac{\eta_t DG\sqrt{N}\lambda^t}{\sqrt{1-\beta_2}} \tag{102}$$

$$\leq f(\mathbf{w}_t) - \eta_t \left( 1 - \frac{G\sqrt{1-\beta_2}}{\epsilon} - \frac{2L\eta_t}{2\epsilon} \right) \sum_{i=1}^{D} \frac{\nabla f(\mathbf{w}_t)_i^2}{\sqrt{\beta_2 \mathbf{v}_{t-1,i}} + \epsilon}$$

$$+ \eta_t \left( \frac{G\sqrt{1-\beta_2}}{\epsilon} + \frac{2L\eta_t}{2\epsilon} \right) \sum_{i=1}^{D} \frac{\sigma_i^2}{\sqrt{\beta_2 \mathbf{v}_{t-1,i}} + \epsilon}$$

$$+ \eta_t \left( \frac{G}{\sqrt{1-\beta_2}} + \frac{LG\eta_t}{2\epsilon\sqrt{1-\beta_2}} \right) Dq_{t-1} + \eta_t \left( \frac{L\eta_t}{2(1-\beta_2)} + \frac{G}{\sqrt{1-\beta_2}} \right) D\sqrt{N}\lambda^t. \tag{103}$$

We adopt some further assumptions from [25], namely that $\eta_t, \beta_2, \epsilon$ are chosen such that

$$\frac{L\eta_t}{2\epsilon} \leq \frac{1}{4}, \qquad \frac{G\sqrt{1-\beta_2}}{\epsilon} \leq \frac{1}{4}. \tag{104}$$

Using these, along with the fact that $\sigma^2 = \sum_i \sigma_i^2$ and $\mathbf{v} \geq 0$ we have that

$$\leq f(\mathbf{w}_t) - \frac{\eta_t}{4} \sum_{i=1}^{D} \frac{\nabla f(\mathbf{w}_t)_i^2}{\sqrt{\beta_2 \mathbf{v}_{t-1,i}} + \epsilon} + \eta_t \left( \frac{G\sqrt{1-\beta_2}}{\epsilon^2} + \frac{L\eta_t}{\epsilon^2} \right) \sigma^2$$

$$+ \eta_t \left( \frac{G}{\sqrt{1-\beta_2}} + \frac{LG\eta_t}{2\epsilon\sqrt{1-\beta_2}} \right) Dq_{t-1} + \eta_t \left( \frac{L\eta_t}{2(1-\beta_2)} + \frac{G}{\sqrt{1-\beta_2}} \right) D\sqrt{N}\lambda^t, \tag{105}$$

and due to our quantization procedure that preserves the maximum value of the second moment, we have that $\mathbf{v}_{t-1,i} \leq G^2$ and therefore

$$\leq f(\mathbf{w}_t) - \frac{\eta_t}{4\left(\sqrt{\beta_2}G + \epsilon\right)} \|\nabla f(\mathbf{w}_t)\|^2 + \eta_t \left( \frac{G\sqrt{1-\beta_2}}{\epsilon^2} + \frac{L\eta_t}{\epsilon^2} \right) \sigma^2$$

$$+ \eta_t \left( \frac{G}{\sqrt{1-\beta_2}} + \frac{LG\eta_t}{2\epsilon\sqrt{1-\beta_2}} \right) Dq_{t-1} + \eta_t \left( \frac{L\eta_t}{2(1-\beta_2)} + \frac{G}{\sqrt{1-\beta_2}} \right) D\sqrt{N}\lambda^t. \tag{106}$$

Therefore, we have that

$$\underset{\boldsymbol{\pi}_t,\mathbf{r}_{t-1},\xi}{\mathbb{E}}[f(\mathbf{w}_{t+1})] - f(\mathbf{w}_t) \leq -\frac{\eta_t}{4\left(\sqrt{\beta_2}G + \epsilon\right)}\|\nabla f(\mathbf{w}_t)\|^2 + \eta_t\left(\frac{G\sqrt{1-\beta_2}}{\epsilon^2} + \frac{L\eta_t}{\epsilon^2}\right)\sigma^2$$

$$+ \eta_t\left(\frac{G}{\sqrt{1-\beta_2}} + \frac{LG\eta_t}{2\epsilon\sqrt{1-\beta_2}}\right)Dq_{t-1}$$

$$+ \eta_t\left(\frac{L\eta_t}{2(1-\beta_2)} + \frac{G}{\sqrt{1-\beta_2}}\right)D\sqrt{N}\lambda^t. \tag{107}$$

Now by setting $\eta_t = \eta$ and taking a telescoping sum we have that

$$\sum_{t=1}^{T}\underset{\boldsymbol{\pi}_t,\mathbf{r}_{t-1},\xi}{\mathbb{E}}[f(\mathbf{w}_{t+1})] - \underset{\boldsymbol{\pi}_{t-1},\mathbf{r}_{t-2},\xi}{\mathbb{E}}[f(\mathbf{w}_t)] \leq -\frac{\eta}{4\left(\sqrt{\beta_2}G + \epsilon\right)}\sum_{t=1}^{T}\|\nabla f(\mathbf{w}_t)\|^2$$

$$+ \eta\left(\frac{G\sqrt{1-\beta_2}}{\epsilon^2} + \frac{L\eta}{\epsilon^2}\right)T\sigma^2 + \eta\left(\frac{G}{\sqrt{1-\beta_2}} + \frac{LG\eta}{2\epsilon\sqrt{1-\beta_2}}\right)D\sum_{t=1}^{T}q_{t-1}$$

$$+ \eta\left(\frac{L\eta}{2(1-\beta_2)} + \frac{G}{\sqrt{1-\beta_2}}\right)D\sqrt{N}\sum_{t=1}^{T}\lambda^t. \tag{108}$$

Rearranging the inequality, doing the cancellations and noting that $f(\mathbf{w}^*) \leq \mathbb{E}_{\boldsymbol{\pi}_T,\mathbf{r}_{T-1},\xi}[f(\mathbf{w}_{T+1})]$
we have that

$$\frac{\eta}{4\left(\sqrt{\beta_2}G + \epsilon\right)}\sum_{t=1}^{T}\|\nabla f(\mathbf{w}_t)\|^2 \leq f(\mathbf{w}_1) - f(\mathbf{w}^*)$$

$$+ \eta\left(\frac{G\sqrt{1-\beta_2}}{\epsilon^2} + \frac{L\eta}{\epsilon^2}\right)T\sigma^2$$

$$+ \eta\left(\frac{G}{\sqrt{1-\beta_2}} + \frac{LG\eta}{2\epsilon\sqrt{1-\beta_2}}\right)D\sum_{t=1}^{T}q_{t-1}$$

$$+ \eta\left(\frac{L\eta}{2(1-\beta_2)} + \frac{G}{\sqrt{1-\beta_2}}\right)D\sqrt{N}\sum_{t=1}^{T}\lambda^t \tag{109}$$

$$\frac{1}{T}\sum_{t=1}^{T}\|\nabla f(\mathbf{w}_t)\|^2 \leq 4(\sqrt{\beta_2}G + \epsilon)\frac{f(\mathbf{w}_1) - f(\mathbf{w}^*)}{\eta T}$$

$$+ 4(\sqrt{\beta_2}G + \epsilon)\left(\frac{G\sqrt{1-\beta_2}}{\epsilon^2} + \frac{L\eta}{\epsilon^2}\right)\sigma^2$$

$$+ 4(\sqrt{\beta_2}G + \epsilon)\left(\frac{G}{\sqrt{1-\beta_2}} + \frac{LG\eta}{2\epsilon\sqrt{1-\beta_2}}\right)D\frac{1}{T}\sum_{t=1}^{T}q_{t-1}$$

$$+ 4(\sqrt{\beta_2}G + \epsilon)\left(\frac{L\eta}{2(1-\beta_2)} + \frac{G}{\sqrt{1-\beta_2}}\right)D\sqrt{N}\frac{1}{T}\sum_{t=1}^{T}\lambda^t \tag{110}$$

$$\frac{1}{T}\sum_{t=1}^{T}\|\nabla f(\mathbf{w}_t)\|^2 \leq 4(\sqrt{\beta_2}G + \epsilon)\frac{f(\mathbf{w}_1) - f(\mathbf{w}^*)}{\eta T}$$

$$+ 4(\sqrt{\beta_2}G + \epsilon)\left(\frac{G\sqrt{1-\beta_2}}{\epsilon^2} + \frac{L\eta}{\epsilon^2}\right)\sigma^2$$

$$+ 4(\sqrt{\beta_2}G + \epsilon)\left(\frac{G}{\sqrt{1-\beta_2}} + \frac{LG\eta}{2\epsilon\sqrt{1-\beta_2}}\right)Dq$$

$$+ 4(\sqrt{\beta_2} + \epsilon)\left(\frac{L\eta}{2(1-\beta_2)} + \frac{G}{\sqrt{1-\beta_2}}\right)D\frac{\lambda\sqrt{N}}{T(1-\lambda)}, \tag{111}$$

with $q = \max_t q_t$. Thus, we end up with something that has similar asymptotic convergence as before, albeit with an additional error term that depends on the quantization procedure of the second moment, *i.e.*,

$$\mathbb{E}[\|\nabla f(\mathbf{w}_a)\|^2] \leq \mathcal{O}\left(\frac{f(\mathbf{w}_1) - f(\mathbf{w}^*)}{\eta T} + \sigma^2 + qDG + \frac{\lambda D \sqrt{N}}{(1-\lambda)T}\right) \tag{112}$$

where $\mathbf{w}_a$ is randomly chosen iterate from $\mathbf{w}_1, \ldots, \mathbf{w}_T$.

### A.4   The case of multiple local updates

It can be beneficial to do multiple local updates at each client, in order to decrease the overall communication cost to reach a specific target accuracy. We will thus extend the convergence proof in order to consider such an approach. Remember that the updates when the second moment is quantized are given as

$$\mathbf{w}_{t+1,i} = \mathbf{w}_{t,i} - \eta_t \frac{\mathbf{g}_{t,i}}{\sqrt{\mathbf{v}_{t,i}} + \epsilon}, \qquad \mathbf{w}_{t+1,i} - \mathbf{w}_{t,i} = -\eta_t \frac{\mathbf{g}_{t,i}}{\sqrt{\mathbf{v}_{t,i}} + \epsilon}, \tag{113}$$

where

$$\mathbf{v}_{t,i} = \beta_2 (\mathbf{v}_{t-1,i} + \mathbf{r}_{t-1,i}) + (1 - \beta_2)\mathbf{g}_{t,i}^2. \tag{114}$$

We will now further assume that each client performs $K$ local updates before it decides whether to send the model to a neighbor. Similarly as before, assume an L-Lipschitz loss function and proceed by:

$$f(\mathbf{w}_{t+1}) \leq f(\mathbf{w}_t) + \nabla f(\mathbf{w}_t)^T (\mathbf{w}_{t+1} - \mathbf{w}_t) + \frac{L}{2}\|\mathbf{w}_{t+1} - \mathbf{w}_t\|^2 \tag{115}$$

$$= f(\mathbf{w}_t) - \eta_t \sum_{i=1}^{D} \nabla f(\mathbf{w}_t)_i \frac{\mathbf{g}_{t,i}}{\sqrt{\mathbf{v}_{t,i}} + \epsilon} + \frac{L\eta_t^2}{2} \sum_{i=1}^{D} \frac{\mathbf{g}_{t,i}^2}{\left(\sqrt{\mathbf{v}_{t,i}} + \epsilon\right)^2}. \tag{116}$$

We will then take an expectation over the randomness at round $t$, given that we are at $\mathbf{w}_t$. We will distinguish two cases; if $t \pmod K$ is zero, then we know that the client will use the randomness of the random walk in order to determine the next node, and if $t \pmod K > 0$, we know that the model will continue staying on this particular client for an update. In the first case, we established the following upper bound

$$\mathbb{E}_{\boldsymbol{\pi}_t, \mathbf{r}_{t-1}, \xi}[f(\mathbf{w}_{t-1})] - f(\mathbf{w}_t) \leq -\frac{\eta_t}{4\left(\sqrt{\beta_2}G + \epsilon\right)}\|\nabla f(\mathbf{w}_t)\|^2 + \eta_t \left(\frac{G\sqrt{1-\beta_2}}{\epsilon^2} + \frac{L\eta_t}{\epsilon^2}\right)\sigma^2$$

$$+ \eta_t \left(\frac{G}{\sqrt{1-\beta_2}} + \frac{LG\eta_t}{2\epsilon\sqrt{1-\beta_2}}\right)Dq_{t-1} + \eta_t \left(\frac{L\eta_t}{2(1-\beta_2)} + \frac{G}{\sqrt{1-\beta_2}}\right)D\sqrt{N}\lambda^t. \tag{117}$$

For the second case, we have to do some more work. We first take the expectation over the randomness of timestep $t$. This specific randomness involves only the data samples of a client $s$, *i.e.*, $\xi_s$ and the quantization noise $\mathbf{r}$. Therefore, we have that

$$\mathbb{E}_{\mathbf{r}_{t-1}, \xi_s}[f(\mathbf{w}_{t+1})] = f(\mathbf{w}_t) - \eta_t \underbrace{\sum_{i=1}^{D} \nabla f(\mathbf{w}_t)_i \mathbb{E}_{\mathbf{r}_{t-1}, \xi_s}\left[\frac{\mathbf{g}_{t,i}}{\sqrt{\mathbf{v}_{t,i}} + \epsilon}\right]}_{T_1}$$

$$+ \frac{L\eta_t^2}{2} \underbrace{\sum_{i=1}^{D} \mathbb{E}_{\mathbf{r}_{t-1}, \xi_s}\left[\frac{\mathbf{g}_{t,i}^2}{\left(\sqrt{\mathbf{v}_{t,i}} + \epsilon\right)^2}\right]}_{T_2}. \tag{118}$$

We then work towards bounding $T_1$. We have that

$$T_1 = -\eta_t \sum_{i=1}^{D} \nabla f(\mathbf{w}_t)_i \underset{\mathbf{r}_{t-1},\xi_s}{\mathbb{E}} \left[ \frac{\mathbf{g}_{t,i}}{\sqrt{\mathbf{v}_{t,i}} + \epsilon} + \frac{\mathbf{g}_{t,i}}{\sqrt{\beta_2 \mathbf{v}_{t-1,i}} + \epsilon} - \frac{\mathbf{g}_{t,i}}{\sqrt{\beta_2 \mathbf{v}_{t-1,i}} + \epsilon} \right.$$
$$\left. + \frac{\nabla f(\mathbf{w}_t)_i}{\sqrt{\beta_2 \mathbf{v}_{t-1,i}} + \epsilon} - \frac{\nabla f(\mathbf{w}_t)_i}{\sqrt{\beta_2 \mathbf{v}_{t-1,i}} + \epsilon} \right] \tag{119}$$

$$= -\eta_t \sum_{i=1}^{D} \frac{\nabla f(\mathbf{w}_t)_i^2}{\sqrt{\beta_2 \mathbf{v}_{t-1,i}} + \epsilon} - \eta_t \sum_{i=1}^{D} \nabla f(\mathbf{w}_t)_i \underset{\mathbf{r}_{t-1},\xi_s}{\mathbb{E}} \left[ \frac{\mathbf{g}_{t,i}}{\sqrt{\mathbf{v}_{t,i}} + \epsilon} - \frac{\mathbf{g}_{t,i}}{\sqrt{\beta_2 \mathbf{v}_{t-1,i}} + \epsilon} \right]$$
$$- \eta_t \sum_{i=1}^{D} \nabla f(\mathbf{w}_t)_i \underset{\xi_s}{\mathbb{E}} \left[ \frac{\mathbf{g}_{t,i}}{\sqrt{\beta_2 \mathbf{v}_{t-1,i}} + \epsilon} - \frac{\nabla f(\mathbf{w}_t)_i}{\sqrt{\beta_2 \mathbf{v}_{t-1,i}} + \epsilon} \right] \tag{120}$$

$$\leq -\eta_t \sum_{i=1}^{D} \frac{\nabla f(\mathbf{w}_t)_i^2}{\sqrt{\beta_2 \mathbf{v}_{t-1,i}} + \epsilon} + \eta_t \sum_{i=1}^{D} |\nabla f(\mathbf{w}_t)_i| \left| \underset{\mathbf{r}_{t-1},\xi_s}{\mathbb{E}} \left[ \frac{\mathbf{g}_{t,i}}{\sqrt{\mathbf{v}_{t,i}} + \epsilon} - \frac{\mathbf{g}_{t,i}}{\sqrt{\beta_2 \mathbf{v}_{t-1,i}} + \epsilon} \right] \right|$$
$$+ \eta_t \sum_{i=1}^{D} \left| \frac{\nabla f(\mathbf{w}_t)_i}{\sqrt{\beta_2 \mathbf{v}_{t-1,i}} + \epsilon} \right| \left| \underset{\xi_s}{\mathbb{E}} \left[ \mathbf{g}_{t,i} - \nabla f(\mathbf{w}_t)_i \right] \right| \tag{121}$$

$$= -\eta_t \sum_{i=1}^{D} \frac{\nabla f(\mathbf{w}_t)_i^2}{\sqrt{\beta_2 \mathbf{v}_{t-1,i}} + \epsilon} + \eta_t \sum_{i=1}^{D} |\nabla f(\mathbf{w}_t)_i| \left| \underset{\mathbf{r}_{t-1},\xi_s}{\mathbb{E}} \left[ \frac{\mathbf{g}_{t,i}}{\sqrt{\mathbf{v}_{t,i}} + \epsilon} - \frac{\mathbf{g}_{t,i}}{\sqrt{\beta_2 \mathbf{v}_{t-1,i}} + \epsilon} \right] \right|$$
$$+ \eta_t \sum_{i=1}^{D} \left| \frac{\nabla f(\mathbf{w}_t)_i}{\sqrt{\beta_2 \mathbf{v}_{t-1,i}} + \epsilon} \right| |\nabla f_s(\mathbf{w}_t)_i - \nabla f(\mathbf{w}_t)_i| , \tag{122}$$

where $\nabla f_s(\mathbf{w}_t)$ denotes the gradient computed on the full dataset of client $s$, *i.e.*, $\mathbb{E}_{\xi_s}[\mathbf{g}]$. From previous derivations and assumptions, we have that

$$\frac{\mathbf{g}_{t,i}}{\sqrt{\mathbf{v}_{t,i}} + \epsilon} - \frac{\mathbf{g}_{t,i}}{\sqrt{\beta_2 \mathbf{v}_{t-1,i}} + \epsilon} \leq \frac{\sqrt{1-\beta_2} \mathbf{g}_{t,i}^2}{\epsilon(\sqrt{\beta_2 \mathbf{v}_{t-1,i}} + \epsilon)} + \frac{q_{t-1}}{\sqrt{1-\beta_2}}, \tag{123}$$

thus, we can continue the upper bound as

$$\leq -\eta_t \sum_{i=1}^{D} \frac{\nabla f(\mathbf{w}_t)_i^2}{\sqrt{\beta_2 \mathbf{v}_{t-1,i}} + \epsilon} + \frac{\eta_t G \sqrt{1-\beta_2}}{\epsilon} \sum_{i=1}^{D} \underset{\xi_s}{\mathbb{E}} \left[ \frac{\mathbf{g}_{t,i}^2}{\sqrt{\beta_2 \mathbf{v}_{t-1,i}} + \epsilon} \right] + \frac{\eta_t D G q_{t-1}}{\sqrt{1-\beta_2}}$$
$$+ \frac{\eta_t G}{\epsilon} \|\nabla f_s(\mathbf{w}_t) - \nabla f(\mathbf{w}_t)\|_1. \tag{124}$$

$$\leq -\eta_t \sum_{i=1}^{D} \frac{\nabla f(\mathbf{w}_t)_i^2}{\sqrt{\beta_2 \mathbf{v}_{t-1,i}} + \epsilon} + \frac{\eta_t G \sqrt{1-\beta_2}}{\epsilon} \sum_{i=1}^{D} \underset{\xi_s}{\mathbb{E}} \left[ \frac{\mathbf{g}_{t,i}^2}{\sqrt{\beta_2 \mathbf{v}_{t-1,i}} + \epsilon} \right] + \frac{\eta_t D G q_{t-1}}{\sqrt{1-\beta_2}}$$
$$+ \frac{\eta_t G \sqrt{D}}{\epsilon} \|\nabla f_s(\mathbf{w}_t) - \nabla f(\mathbf{w}_t)\|_2. \tag{125}$$

$$\leq -\eta_t \sum_{i=1}^{D} \frac{\nabla f(\mathbf{w}_t)_i^2}{\sqrt{\beta_2 \mathbf{v}_{t-1,i}} + \epsilon} + \frac{\eta_t G \sqrt{1-\beta_2}}{\epsilon} \sum_{i=1}^{D} \underset{\xi_s}{\mathbb{E}} \left[ \frac{\mathbf{g}_{t,i}^2}{\sqrt{\beta_2 \mathbf{v}_{t-1,i}} + \epsilon} \right] + \frac{\eta_t D G q_{t-1}}{\sqrt{1-\beta_2}}$$
$$+ \frac{\eta_t \sqrt{D} G^2}{2\epsilon} + \frac{\eta_t \sqrt{D}}{2\epsilon} \|\nabla f_s(\mathbf{w}_t) - \nabla f(\mathbf{w}_t)\|_2^2, \tag{126}$$

where the last step is due to Young's inequality applied to $G\|\nabla f_s(\mathbf{w}_t) - \nabla f(\mathbf{w}_t)\|_2$.

By then using two of our assumptions, namely, the bounded (local) variance and bounded difference between the local and global gradients, we have that

$$T_1 \leq -\eta_t \sum_{i=1}^{D} \frac{\nabla f(\mathbf{w}_t)_i^2}{\sqrt{\beta_2 \mathbf{v}_{t-1,i}} + \epsilon} + \frac{\eta_t G \sqrt{1-\beta_2}}{\epsilon} \sum_{i=1}^{D} \underset{\xi_s}{\mathbb{E}} \left[ \frac{\mathbf{g}_{t,i}^2}{\sqrt{\beta_2 \mathbf{v}_{t-1,i}} + \epsilon} \right]$$
$$+ \frac{\eta_t D G q_{t-1}}{\sqrt{1-\beta_2}} + \frac{\eta_t \sqrt{D} G^2}{2\epsilon} + \frac{\eta_t \sqrt{D} \zeta^2}{2\epsilon}. \tag{127}$$

By reusing previous derivations, we can also show that

$$T_2 \le \frac{2L\eta_t^2}{2\epsilon} \sum_{i=1}^{D} \mathbb{E}_{\xi_s} \left[ \frac{\mathbf{g}_{t,i}^2}{\sqrt{\beta_2 \mathbf{v}_{t-1,i}} + \epsilon} \right] + \frac{LDGq_{t-1}\eta_t^2}{2\epsilon\sqrt{1-\beta_2}}. \tag{128}$$

Thus, by putting everything together,

$$\begin{aligned}
\mathbb{E}_{\mathbf{r}_{t-1},\xi_s} [f(\mathbf{w}_{t+1})] &\le f(\mathbf{w}_t) - \eta_t \sum_{i=1}^{D} \frac{\nabla f(\mathbf{w}_t)_i^2}{\sqrt{\beta_2 \mathbf{v}_{t-1,i}} + \epsilon} \\
&+ \left( \frac{\eta_t G\sqrt{1-\beta_2}}{\epsilon} + \frac{L\eta_t^2}{\epsilon} \right) \sum_{i=1}^{D} \underbrace{\mathbb{E}_{\xi_s} \left[ \frac{\mathbf{g}_{t,i}^2}{\sqrt{\beta_2 \mathbf{v}_{t-1,i}} + \epsilon} \right]}_{T_3} \\
&+ \left( \frac{\eta_t}{\sqrt{1-\beta_2}} + \frac{L\eta_t^2}{2\epsilon\sqrt{1-\beta_2}} \right) DGq_{t-1} + \frac{\eta_t\sqrt{D}}{2\epsilon} \left( G^2 + \zeta^2 \right). \tag{129}
\end{aligned}$$

We will now bound $T_3$.

$$T_3 = \mathbb{E}_{\xi_s} \left[ \frac{\mathbf{g}_{t,i}^2}{\sqrt{\beta_2 \mathbf{v}_{t-1,i}} + \epsilon} \right] \le \frac{\nabla f_s(\mathbf{w}_t)^2 + \sigma_{li}^2}{\sqrt{\beta_2 \mathbf{v}_{t-1,i}} + \epsilon} \tag{130}$$

$$= \frac{(\nabla f_s(\mathbf{w}_t) - \nabla f(\mathbf{w}_t) + \nabla f(\mathbf{w}_t))^2 + \sigma_{li}^2}{\sqrt{\beta_2 \mathbf{v}_{t-1,i}} + \epsilon} \tag{131}$$

$$\le \frac{2}{\sqrt{\beta_2 \mathbf{v}_{t-1,i}} + \epsilon} \left( \frac{\sigma_{li}^2}{2} + (\nabla f_s(\mathbf{w}_t)_i - \nabla f(\mathbf{w}_t)_i)^2 + \nabla f(\mathbf{w}_t)_i^2 \right). \tag{132}$$

Therefore, we have that

$$\begin{aligned}
\mathbb{E}_{\mathbf{r}_{t-1},\xi_s} [f(\mathbf{w}_{t+1})] &\le f(\mathbf{w}_t) - \eta_t \sum_{i=1}^{D} \frac{\nabla f(\mathbf{w}_t)_i^2}{\sqrt{\beta_2 \mathbf{v}_{t-1,i}} + \epsilon} \\
&+ 2 \left( \frac{\eta_t G\sqrt{1-\beta_2}}{\epsilon} + \frac{L\eta_t^2}{\epsilon} \right) \sum_{i=1}^{D} \frac{\nabla f(\mathbf{w}_t)_i^2}{\sqrt{\beta_2 \mathbf{v}_{t-1,i}} + \epsilon} \\
&+ 2 \left( \frac{\eta_t G\sqrt{1-\beta_2}}{\epsilon^2} + \frac{L\eta_t^2}{\epsilon^2} \right) \left( \frac{\sigma_l^2}{2} + \|\nabla f_s(\mathbf{w}_t) - \nabla f(\mathbf{w}_t)\|_2^2 \right) \\
&+ \left( \frac{\eta_t}{\sqrt{1-\beta_2}} + \frac{L\eta_t^2}{2\epsilon\sqrt{1-\beta_2}} \right) DGq_{t-1} + \frac{\eta_t\sqrt{D}}{2\epsilon} \left( G^2 + \zeta^2 \right) \tag{133} \\
&\le f(\mathbf{w}_t) - \left( \eta_t - \frac{2\eta_t G\sqrt{1-\beta_2}}{\epsilon} - \frac{2L\eta_t^2}{\epsilon} \right) \sum_{i=1}^{D} \frac{\nabla f(\mathbf{w}_t)_i^2}{\sqrt{\beta_2 \mathbf{v}_{t-1,i}} + \epsilon} \\
&+ 2 \left( \frac{\eta_t G\sqrt{1-\beta_2}}{\epsilon^2} + \frac{L\eta_t^2}{\epsilon^2} \right) \left( \frac{\sigma_l^2}{2} + \zeta^2 \right) \\
&+ \left( \frac{\eta_t}{\sqrt{1-\beta_2}} + \frac{L\eta_t^2}{2\epsilon\sqrt{1-\beta_2}} \right) DGq_{t-1} + \frac{\eta_t\sqrt{D}}{2\epsilon} \left( G^2 + \zeta^2 \right). \tag{134}
\end{aligned}$$

In order to maintain an upper bound, we have to impose some further assumptions, namely that

$$\frac{2L\eta_t}{\epsilon} \le \frac{1}{2}, \qquad \frac{2G\sqrt{1-\beta_2}}{\epsilon} \le \frac{1}{4}. \tag{135}$$

In this way, we have that

$$\underset{\mathbf{r}_{t-1},\xi_s}{\mathbb{E}}[f(\mathbf{w}_{t+1})] \le f(\mathbf{w}_t) - \frac{\eta_t}{4} \sum_{i=1}^{D} \frac{\nabla f(\mathbf{w}_t)_i^2}{\sqrt{\beta_2 \mathbf{v}_{t-1,i}} + \epsilon}$$

$$+ 2\left( \frac{\eta_t G \sqrt{1-\beta_2}}{\epsilon^2} + \frac{L\eta_t^2}{\epsilon^2} \right) \left( \frac{\sigma_l^2}{2} + \zeta^2 \right)$$

$$+ \left( \frac{\eta_t}{\sqrt{1-\beta_2}} + \frac{L\eta_t^2}{2\epsilon\sqrt{1-\beta_2}} \right) DG q_{t-1} + \frac{\eta_t \sqrt{D}}{2\epsilon} \left( G^2 + \zeta^2 \right)$$

$$(136)$$

$$\underset{\mathbf{r}_{t-1},\xi_s}{\mathbb{E}}[f(\mathbf{w}_{t+1})] - f(\mathbf{w}_t) \le -\frac{\eta_t}{4\left(\sqrt{\beta_2}G + \epsilon\right)} \|\nabla f(\mathbf{w}_t)\|^2$$

$$+ 2\left( \frac{\eta_t G \sqrt{1-\beta_2}}{\epsilon^2} + \frac{L\eta_t^2}{\epsilon^2} \right) \left( \frac{\sigma_l^2}{2} + \zeta^2 \right)$$

$$+ \left( \frac{\eta_t}{\sqrt{1-\beta_2}} + \frac{L\eta_t^2}{2\epsilon\sqrt{1-\beta_2}} \right) DG q_{t-1} + \frac{\eta_t \sqrt{D}}{2\epsilon} \left( G^2 + \zeta^2 \right),$$

$$(137)$$

which constitutes our final bound for the case of a client locally updating the model without doing the random walk. We now assume $\eta_t = \eta, q = \max_t q_t$ and take a telescoping sum while considering both of these cases, where $\mathbb{E}_t$ corresponds to an expectation over all randomness at round $t$.

$$\sum_{t=1}^{KT} \underset{t}{\mathbb{E}}[f(\mathbf{w}_{t+1})] - \underset{t-1}{\mathbb{E}}[f(\mathbf{w}_t)] \le -\frac{\eta}{4\left(\sqrt{\beta_2}G + \epsilon\right)} \sum_{t(\bmod K)=0} \|\nabla f(\mathbf{w}_t)\|^2$$

$$+ \eta T \left( \frac{G\sqrt{1-\beta_2}}{\epsilon^2} + \frac{L\eta}{\epsilon^2} \right) \sigma^2$$

$$+ \eta T \left( \frac{G}{\sqrt{1-\beta_2}} + \frac{LG\eta}{2\epsilon\sqrt{1-\beta_2}} \right) Dq$$

$$+ \eta \left( \frac{L\eta}{2(1-\beta_2)} + \frac{G}{\sqrt{1-\beta_2}} \right) D\sqrt{N} \sum_{i=1}^{T} \lambda^i$$

$$- \frac{\eta}{4\left(\sqrt{\beta_2}G + \epsilon\right)} \sum_{t(\bmod K)\neq 0} \|\nabla f(\mathbf{w}_t)\|^2$$

$$+ 2T(K-1)\left( \frac{\eta G\sqrt{1-\beta_2}}{\epsilon^2} + \frac{L\eta^2}{\epsilon^2} \right) \left( \frac{\sigma_l^2}{2} + \zeta^2 \right)$$

$$+ T(K-1)\left( \frac{\eta}{\sqrt{1-\beta_2}} + \frac{L\eta^2}{2\epsilon\sqrt{1-\beta_2}} \right) DG q + \frac{T(K-1)\eta\sqrt{D}}{2\epsilon} \left( G^2 + \zeta^2 \right) \quad (138)$$

$$\le -\frac{\eta}{4\left(\sqrt{\beta_2}G + \epsilon\right)} \sum_{t=1}^{T} \|\nabla f(\mathbf{w}_t)\|^2$$

$$+ \eta T \left( \frac{G\sqrt{1-\beta_2}}{\epsilon^2} + \frac{L\eta}{\epsilon^2} \right) \sigma^2 + \frac{T(K-1)\eta\sqrt{D}}{2\epsilon} \left( G^2 + \zeta^2 \right)$$

$$+ 2T(K-1)\left( \frac{\eta G\sqrt{1-\beta_2}}{\epsilon^2} + \frac{L\eta^2}{\epsilon^2} \right) \left( \frac{\sigma_l^2}{2} + \zeta^2 \right)$$

$$+ \eta KT \left( \frac{1}{\sqrt{1-\beta_2}} + \frac{L\eta}{2\epsilon\sqrt{1-\beta_2}} \right) DG q$$

$$+ \eta \left( \frac{L\eta}{2(1-\beta_2)} + \frac{G}{\sqrt{1-\beta_2}} \right) D\frac{\lambda\sqrt{N}}{1-\lambda}. \quad (139)$$

Therefore, by simplifying and re-arranging we have that

$$\frac{\eta}{4\left(\sqrt{\beta_2}G+\epsilon\right)}\sum_{t=1}^{KT}\|\nabla f(\mathbf{w}_t)\|^2 \leq f(\mathbf{w}_1)-f(\mathbf{w}^*)+\eta T\left(\frac{G\sqrt{1-\beta_2}}{\epsilon^2}+\frac{L\eta}{\epsilon^2}\right)\sigma^2$$
$$+\frac{T(K-1)\eta\sqrt{D}}{2\epsilon}\left(G^2+\zeta^2\right)$$
$$+2T(K-1)\left(\frac{\eta G\sqrt{1-\beta_2}}{\epsilon^2}+\frac{L\eta^2}{\epsilon^2}\right)\left(\frac{\sigma_l^2}{2}+\zeta^2\right)$$
$$+\eta KT\left(\frac{1}{\sqrt{1-\beta_2}}+\frac{L\eta}{2\epsilon\sqrt{1-\beta_2}}\right)DGq$$
$$+\eta\left(\frac{L\eta}{2(1-\beta_2)}+\frac{G}{\sqrt{1-\beta_2}}\right)D\frac{\lambda\sqrt{N}}{1-\lambda} \tag{140}$$

$$\frac{1}{KT}\sum_{t=1}^{KT}\|\nabla f(\mathbf{w}_t)\|^2 \leq 4(\sqrt{\beta_2}G+\epsilon)\frac{f(\mathbf{w}_1)-f(\mathbf{w}^*)}{\eta KT}$$
$$+4(\sqrt{\beta_2}G+\epsilon)\left(\frac{G\sqrt{1-\beta_2}}{\epsilon^2}+\frac{L\eta}{\epsilon^2}\right)\frac{\sigma^2}{K}$$
$$+4(\sqrt{\beta_2}G+\epsilon)\frac{(K-1)\sqrt{D}}{2K\epsilon}\left(G^2+\zeta^2\right)$$
$$+8(\sqrt{\beta_2}G+\epsilon)\frac{K-1}{K}\left(\frac{G\sqrt{1-\beta_2}}{\epsilon^2}+\frac{L\eta}{\epsilon^2}\right)\left(\frac{\sigma_l^2}{2}+\zeta^2\right)$$
$$+4(\sqrt{\beta_2}G+\epsilon)\left(\frac{1}{\sqrt{1-\beta_2}}+\frac{L\eta}{2\epsilon\sqrt{1-\beta_2}}\right)DGq$$
$$+4(\sqrt{\beta_2}G+\epsilon)\left(\frac{L\eta}{2(1-\beta_2)}+\frac{G}{\sqrt{1-\beta_2}}\right)D\frac{\lambda\sqrt{N}}{(1-\lambda)KT}. \tag{141}$$

We thus end up with

$$\mathbb{E}[\|\nabla f(\mathbf{w}_a)\|^2] \leq \mathcal{O}\left(\frac{f(\mathbf{w}_1)-f(\mathbf{w}^*)}{\eta KT}+\frac{\sigma^2}{K}+qDG+\frac{K-1}{K}(\sigma_l^2+\zeta^2+G^2)+\frac{\lambda D\sqrt{N}}{(1-\lambda)KT}\right), \tag{142}$$

where $\mathbf{w}_a$ is randomly chosen iterate from $\mathbf{w}_1,\ldots,\mathbf{w}_{KT}$.

## B  Experimental details

The experiments in this paper were performed on workstations, each of which is equipped with a single Nvidia RTX 2080Ti GPU, as well as on machines equipped with Nvidia V100 and A100 GPUs. All experiments where performed on single GPUs only.

### B.1  Models and datasets

**FEMNIST**  FEMNIST is based on the extended MNIST (EMNIST) dataset [27] and is adapted to the federated setting. Its images are MNIST-like and consist of 63 different classes of handwritten letters and digits. In the federated setting, we consider all data-point written by an individual person to constitute an individual client's dataset. Any non-i.i.d. ness therefore stems from the different writing-styles, as well as from the varying size of the clients' datasets. Additionally, the size of the individual clients' datasets differ significantly. We experiment with the dataset that is originally published by [26] and use their provided code to generate the dataset. We remark that the statistics reported by the original creators do not align with the result of the provided code when we run it. In Table 3, we report the dataset's statistic as we compute them. We use the standard LeNet-5 convolutional network [28] in all FEMNIST experiments.

Table 3: Statistics of federated training sets

| Dataset | Number of clients | Total samples | Samples per device mean | std |
|---------|-------------------|---------------|-------------------------|-----|
| FEMNIST | 3597 | $734,463$ | 204.19 | 79.95 |
| Shakespeare | 660 | $3,678,451$ | 5573.41 | 6460.77 |
| StackOverflow | $342,477$ | $135,818,730$ | 396.58 | 1278.94 |
| Cifar10 | 100 | $45,000$ | 450 | 102 |
| Cifar100 | 500 | $50,000$ | 100 | 0 |

**Shakespeare**   We follow [26] in the creation of the Shakespeare dataset. Each unique character's spoken lines across all of Shakespear's plays constitutes the dataset for an individual client in the federated setup. This dataset is naturally non-i.i.d due to the 660 different characters' identities and corresponding spoken lines. Our model architecture consists of the 2-layer LSTM model proposed in [26]. The task is formulated as next-character-prediction task over an alphabet of 80 characters. The cross-entropy term is computed only on the next character prediction based on an encoding of a 80 character long sentences. For the statistics in Table 3, we report each pair of 80 characters-long sentences plus next character as a single sample. Please note that the differences between our statistics and what is reported by [26] has been raised in a github issue[2].

**StackOverflow**   The StackOverflow dataset [32] is a collection of questions and answers scraped from the StackOverflow website. Each user of that website who posted there within a specific time-frame is a client and their posts constitute its dataset. In this work, we compare on the tag prediction task described in [16] using a simple logistic regression model. Each posting is tagged with one or multiple of 500 possible tags, such that a client performs one-vs-all classification across these 500 tags. Each posting is represented as a normalized bag-of-words across the $10,000$ most frequent words in the training set. Since the validation-set is very large, we report learning curves by sampling in the main paper were created by selecting the first $10\%$ of data-points. For the results in Table 1, the final model was evaluated on the test set. We do 30k steps in the case of random walk optimization and 1.5k steps for FedAvg.

**CIFAR10/CIFAR100**   For the CIFAR10 dataset, we split the $45k$ training examples across $100$ clients in a non-i.i.d. way according to the label-skew setting presented in [16] with $\alpha = 1.0$. For CIFAR100, the training set is split across 500 clients in a non-i.i.d. way following the PAM method in [16]. We keep $\alpha = 1.0, \beta = 10.0$. Experiments on both datasets are performed with the standard ResNet-20 architecture for CIFAR data-sets, albeit with BatchNorm replaced by GroupNorm with 2 groups.

**Gossip Averaging**   For gossip averaging experiments, we use CIFAR10 and Shakespeare. The data split for both tasks is equivalent to the one described above, and so are the models. We also use the same small-world communication graph with the average degree of 5 and $\beta = 0.5$. Training is done similarly to prior work on gossiping (e.g., [21]): using gradient descent with momentum (0.9) and weight decay ($10^{-4}$); learning rate is 0.1. In CIFAR10 experiment, all clients are updated in each round, but in Shakespeare, due to a significant slow-down caused by the larger federation and model, only 10 users get updated in every round. The latter, however, doesn't appear to impact the accuracy-communication trade-off, and thus our conclusions. Finally, the PowerGossip communication savings are accounted by applying the compression ratio achieved in [21] for CIFAR10 (as the model is the same, ResNet-20), and by computing the ratio directly, layer-by-layer, following the PowerGossip method with 1 iteration for Shakespeare.

## B.2   Hyperparameters

For all of the tasks involving a neural network (*i.e.*, all tasks we considered, except StackOverflow) for RW-Adam we used the default Adam hyperparameters (without momentum), *i.e.*, a learning rate of $1e - 3$, $\beta_2 = 0.999$, $\beta_1 = 0$ and $\epsilon = 1e - 7$. For the quantized version of Adam, we used either 4 or 5 bits per parameter, determined by comparing validation performance against the unquantized

---

[2]`https://github.com/TalwalkarLab/leaf/issues/13`

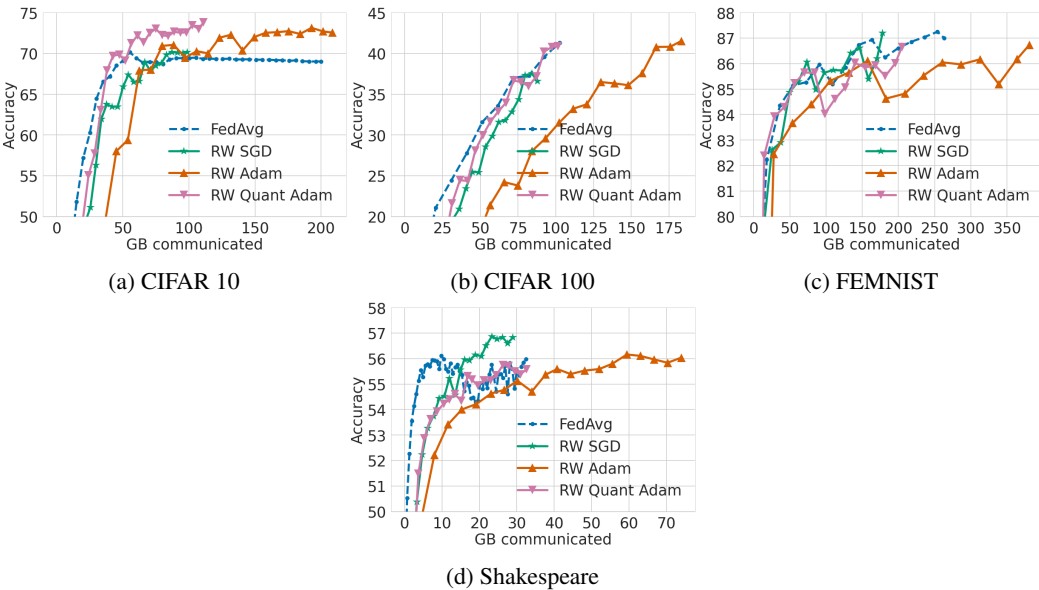

(a) CIFAR 10       (b) CIFAR 100       (c) FEMNIST

(d) Shakespeare

Figure 4: Average (over different random seeds) learning curves of validation accuracy as a function of the cumulative communication costs.

Table 4: Average (over different random seeds) test-set accuracy at the end of training and total communication (in GB) along with the standard error.

| Method | CIFAR10 $K = 1, b = 4$ | | CIFAR100 $K = 1, b = 5$ | | FEMNIST $K = 1, b = 5$ | | Shakespeare $K = 10, b = 4$ | |
|---|---|---|---|---|---|---|---|---|
| | Acc. | Comm. | Acc. | Comm. | Acc. | Comm. | Acc. | Comm. |
| FedAvg | $69.0 \pm 0.4$ | 120 | $41.3 \pm 0.5$ | 102 | $87.0 \pm 0.3$ | 263 | $56.0 \pm 0.2$ | 33 |
| RW SGD | $70.2 \pm 1.3$ | 99 | $36.6 \pm 0.7$ | 86 | $87.2 \pm 0.1$ | 177 | $56.8 \pm 0.1$ | 28 |
| RW Adam | $72.6 \pm 0.2$ | 211 | $41.5 \pm 0.3$ | 180 | $86.7 \pm 0.1$ | 302 | $56.0 \pm 0.1$ | 58 |
| RW QAdam | $73.8 \pm 0.3$ | 112 | $40.9 \pm 0.7$ | 100 | $86.7 \pm 0.1$ | 163 | $55.6 \pm 0.1$ | 31 |

version of Adam. For RW-SGD we tuned the learning rate on a per-experiment basis via a validation set. The range we considered was $[1e-3, 1]$. We found that a learning rate of $1e-1$ worked well on all of the vision tasks, whereas a learning rate of $1$ worked well for Shakespeare. For StackOverflow, we found that we need much larger learning rates; for RW-Adam and RW-QAdam we used a learning rate of $1e-1$ whereas for SGD we used a learning rate of $1e5$. The number of local steps in the case of random-walk optimization was picked from $\{1, 3, 5, 10\}$, again determined on a validation set. We used a batch-size of $128$ for CIFAR10/100, Shakespeare, FEMNIST, and a batch-size of $100$ for StackOverflow.

For FedAvg, we used the hyperparameters provided at [16] for StackOverflow, whereas for CIFAR 10 / CIFAR100 / FEMNIST, we used SGD locally with a learning rate of $0.05$, $1$ local epoch and Adam with the default hyperparameters at the server (with momentum), *i.e.*, a learning rate of $1e-3$, $\beta_1 = 0.9$, $\beta_2 = 0.999$ and $\epsilon = 1e-7$. For Shakespeare we found it beneficial to use locally SGD with a learning rate of $1.0$, $1$ local epoch and similarly SGD with a learning rate of $1.0$ at the server. For CIFAR10/CIFAR100/FEMNIST/StackOverflow we sampled $10$ clients per round, whereas we sampled $66$ clients per round for Shakespeare.

## C   Additional results

In this section we provide an estimate of the variability of our results according to different random seeds on CIFAR10, CIFAR100, FEMNIST and Shakespeare; we used $4$ random seeds for CIFAR 10, Shakespeare and $3$ random seeds for FEMNIST and CIFAR100. In Figure 4 and Table 4 we

provide the average learning curves and the average test-set accuracy along with its standard error respectively. We can see that, overall, our results have similar variability to the ones from FedAvg and do not change our conclusions; we can get similar or better accuracy to FedAvg with comparable communication costs. Note that for CIFAR100, the performance of FedAvg is lower than what we reported in the main text; this is due to choosing a point in training for FedAvg that has similar communication costs as our RW-QAdam.

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
