# OpenReview forum: "Decentralized Learning with Random Walks and Communication-Efficient Adaptive Optimization"
_NeurIPS.cc/2022/Workshop/Federated_Learning — FL-NeurIPS 2022 Poster_

### Official Review · Reviewer_exhX · 2022-10-15
**Review for Decentralized Learning with Random Walks and Communication-Efficient Adaptive Optimization**

This paper develops a novel decentralized learning algorithm by replacing gossip averaging with random walks as the communication protocol. Specifically, the authors designed the protocol to allow the communication to happen between only a pair of agents. The authors also theoretically analyzed three scenarios: random walk with Adam, random walk with Adam and second moment quantization, and random walk with Adam, second moment quantization and multiple local updates. The quantization and multiple local updates were used to further reduce the communication when second moment was transmitted. The authors showed the sublinear convergence rates for all three different cases. To validated the proposed algorithms, the authors leveraged multiple benchmark datasets to show the comparative performance as the FL algorithm without communication overhead.

The investigated topic in this paper is interesting as decentralized learning has been quite popular in the era of big data and communication overhead still remains a challenge. The paper is also easy to follow and well written. While in my perspective, the theoretical contribution is marginal as the algorithm just combined different existing techniques together, such as random walk, communication compression by quantization, multiple local updates, and Adam. It is fine if such a combination could reveal new and solid results, but the convergence rate is  still the same as that of existing works and the empirical results are not promising. Though the authors have utilized different tasks and models. The proposed algorithm is not that competitive to traditional FedAvg based on the plots. Would it because the usage of random walk protocol? Intuitively, though random walk protocol is able to reduce the communication, it may hurt the optimality due to the limited information learned from other agents in the network. I think the authors should analyze the tradeoff between communication and optimality as the latter would affect the model performance.

Another weakness in the paper is that the baseline method. The authors only leveraged Gossip Averaging in the main contents and FedAvg in the appendix. I believe there have been quite a few other communication-efficient decentralized learning proposed in literature. The authors should at least adopt one or two recently approaches to showcase the overperforming capability of the proposed algorithms.

---

### Official Review · Reviewer_k8DE · 2022-10-16

The authors study the communication bottleneck in federated learning (FL) when there is no central server and clients perform peer-to-peer communication to transmit updated models. Their main contribution is to provide an adaptive optimization strategy under this scenario while reducing the communication cost by (1) compressing the state parameters and (2) by performing several local updates in between communication as in FedAvg. The authors show that the proposed method is promising in comparison to the baselines on CIFAR-10, CIFAR-100, FEMNIST, Shakespeare, and StackOverflow datasets.

The paper is theoretically sound and has interesting results. The way of compressing the state variables and reducing the number of communication rounds via local training are two pretty standard strategies, but it is still valuable to see that they work well under the proposed random-walk strategy as well. Moreover, the authors provide theoretical guarantees on convergence, which makes their accuracy-communication analysis more rigorous.

---

### Official Review · Reviewer_XL84 · 2022-10-18
**good idea, need more results and comparisons**

In this paper, the authors propose to use a random walk-based decentralized learning algorithm. As the authors have claimed and cited, this idea is not novel, and there is some preliminary work on this approach. So, the actual contribution is in terms of actual empirical results and some more analysis. There are not many comparisons, and there are so many things added like compression and multiple local updates which confuse the results to understand what contributes what. Further, a proper systematic ablation study of different parameters and different topologies of the graph would be good. For example, I am very curious to see what happens for this algorithm when you are working on a ring topology. Does the directionality of the walk matter? is the sampling with replacement or without replacement?

Overall, good work but needs more clarification.

---

### Decision · Program_Chairs · 2022-10-20

Accept (Poster)